# Spatial and Temporal Study of Supernatant Process Water Pond in Tailings Storage Facilities: Use of Remote Sensing Techniques for Preventing Mine Tailings Dam Failures

Carlos Cacciuttolo [1],*  and Deyvis Cano [2]

1   Civil Works and Geology Department, Catholic University of Temuco, Temuco 4780000, Chile
2   Programa Académico de Ingeniería Ambiental, Universidad de Huánuco, Huánuco 10001, Peru
*   Correspondence: ccacciuttolo@uct.cl or carlos.cacciuttolo@gmail.com

**Abstract:** Considering the global impact on society due to tailings storage facilities (TSFs) accidents, this article describes a study to monitor mine tailings management and prevent mining tailings dam failures, considering the analysis of different TSFs real cases. The spatial and temporal dynamic behavior of the supernatant process water pond of the TSFs is studied as a critical issue, using remote sensing techniques based on multispectral satellite imagery. To understand the current state of the art, a brief description of engineering studies for the control and management of the supernatant process water pond in TSFs is presented. This research considers the main method of the study of practical cases with the use of techniques of multispectral interpretation of satellite images from the Sentinel 2 remote sensor. In addition, the management of tools such as Geographical Information System (GIS) and Google Earth Engine (GEE) is implemented, as well as the application of some spectral indices such as NDWI and the joint use of (i) NDVI, (ii) mNDWI, and (iii) EVI. Real TSF cases are analyzed, including the dam failures of Jagersfontain TSF in South Africa and Williamson TSF in Tanzania. Finally, this article concludes that the size, location, and temporal variability of the supernatant process water pond within a TSF has a direct impact on safety and the possible potential risk of the physical instability of tailings dams.

**Keywords:** tailings storage facility; remote sensing techniques; multispectral satellite imagery; Google Earth Engine; NDWI; NDVI; mNDWI; EVI

## 1. Introduction

### 1.1. Responsible and Safe Mine Tailings Management—A Worldwide Priority

The generation of mine tailings worldwide is reaching unprecedented productions, reaching levels of generation per year equivalent to 14 billion metric tons [1]. Unfortunately, in recent decades, serious tailings dam rupture accidents have occurred that have negatively impacted the environment and the communities that live in the vicinity of the tailings storage facilities (TSFs) [2–10]. Some historical examples are (i) Los Frailes, Spain, 1998, (ii) Baia Mare, Romania, 2000 (iii) Kolontar, Hungary, 2010, (iv) Mount Polley, Canada, 2014, (v) Fundao Samarco, Brazil, 2015, (vi) Brumadinho, Brazil, 2019, (vii) Jagersfontain, South Africa, 2022, and (viii) Williamson, Tanzania, 2022 [11]. These events have generated environmental contamination, injured people, and in some cases resulted in the death of people, which has caused a call to attention for society in general about the level of risk that TSFs represent worldwide [12,13]. This has damaged the reputation of mining companies in general with neighboring communities, provoking a reaction from mining investors to improve tailings management practices, raise safety standards, and implement the best available technologies (BATs) [14–16].

According to the different analyses and investigations carried out to determine the causes of TSF failures worldwide, it has been identified that one of the most critical aspects in the management of tailings deposits is the control of the water stored inside the TSF

reservoir area, either process water from tailings sedimentation or precipitation water (Figure 1) [17,18]. Along with this, the complexity of the interaction and multidisciplinary coordination of professionals (e.g., engineering companies, construction companies, and mining companies, among others) in the operation over time (years) of the TSF has been identified as a potential cause of errors or nonconformities in the performance of good tailings management practices [15]. The human factor has also been identified as a key aspect in TSF failures considering manual monitoring, revealing the lack of intelligent monitoring systems in real time of key critical parameters to detect anomalies in the behavior of the physical stability of TSFs [1,19,20].

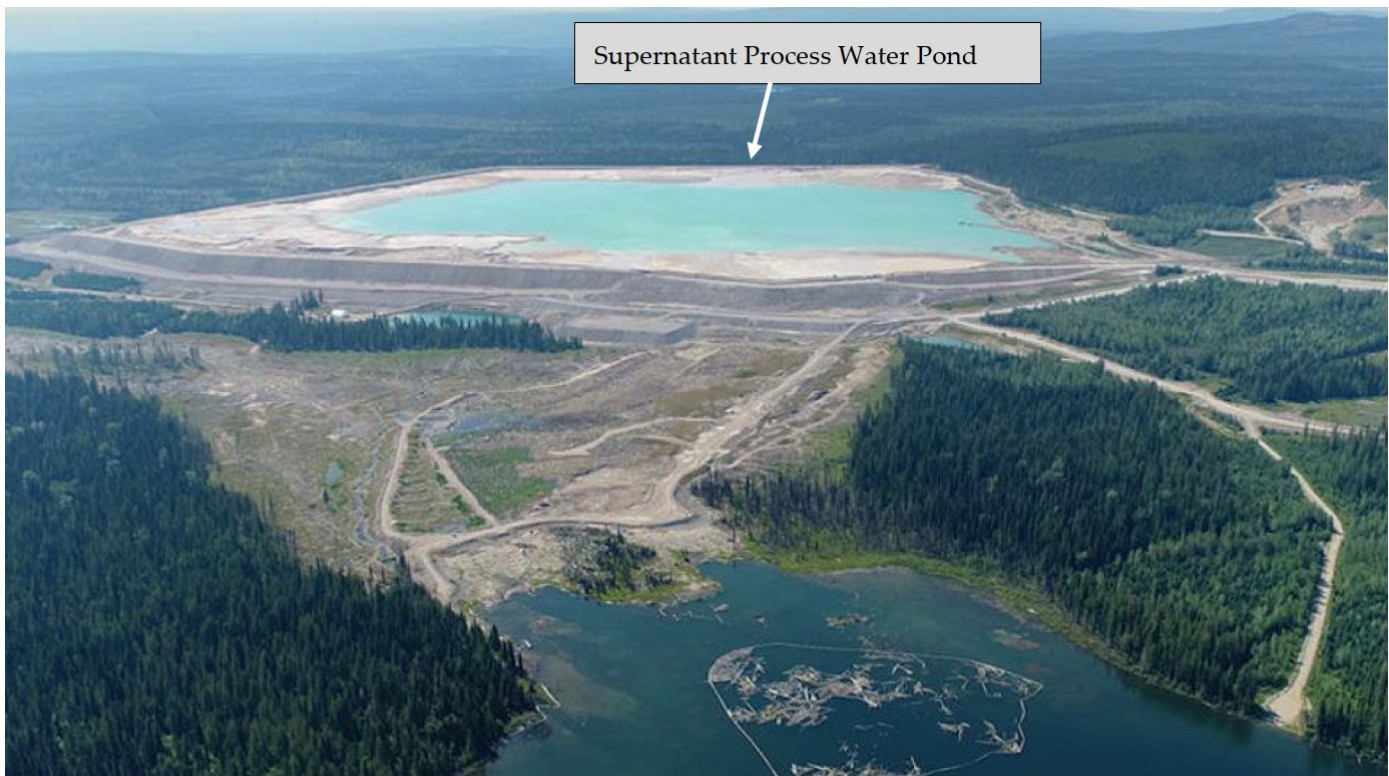

**Figure 1.** Mount Polley TSF—Canada, before the dam failure in 2014. The presence of a huge supernatant process water pond.

Unfortunately, mining does not have a history of environmental care; its history of depositing mine tailings in rivers and the sea has caused distrust and fear on the part of many communities, since many of these mine tailings remain currently as environmental liabilities without any remediation [15]. Some of these cases are (i) Chañaral Bay, Chile, (ii) Ite Bay, Peru, and (iii) Portman Bay, Spain. These cases of old mining where there was no adequate environmental regulation, as there is today, still impact public opinion and local communities [21].

All of the aforementioned have produced a movement in society that demands safe and responsible management of mine tailings by mining companies, demanding (i) greater economic investments in the control of mine tailings, (ii) use of highly qualified personnel for the design, construction, management, and administration of tailings deposits, (iii) intelligent monitoring with sensors based on data mining, artificial intelligence, machine learning, and Internet of Things (IoT) implemented 24 h a day, 365 days a year, and (iv) the use of the best available technologies (BATs) such as thickened tailings (TTD), paste tailings (PTD), and filtered tailings (FTD) [11,22–26].

This has revealed a new era of determining how to manage mine tailings by mining companies, being that today mine tailings are not mining waste but a key asset and protago-

nist within the heart of mining investment, which if not properly managed can cause paralysis of a mining operation and rejection by the community and local authorities [6,26–31].

### 1.2. Aim of the Article

Considering the global impact on society due to tailings deposit failures, this article's main objectives are as follows:

- To describe a study of practical cases of mine tailings management to prevent TSF failures. This research considers different cases of tailings deposits, where the spatial and temporal behavior of the supernatant process water pond and wet tailings zone of the deposited mine tailings is analyzed, using remote sensing techniques based on multispectral satellite imagery.
- In addition, as a second objective to understand the current state of the art, a brief description of the theory (engineering studies) and practice (construction and operation activities) for the control and management of the supernatant process water pond in tailings deposits are presented.

As a methodology, this research considers the study of practical cases with the use of techniques for management and multispectral interpretation of Sentinel 2 satellite images with the use of spectral indices such as NDWI (Normalized Difference Water Index) and the joint use of (i) NDVI (Normalized Difference Vegetation Index), (ii) mNDWI (Modified Normalized Difference Water Index), and (iii) EVI (Enhanced Vegetation Index) (see Figure 2).

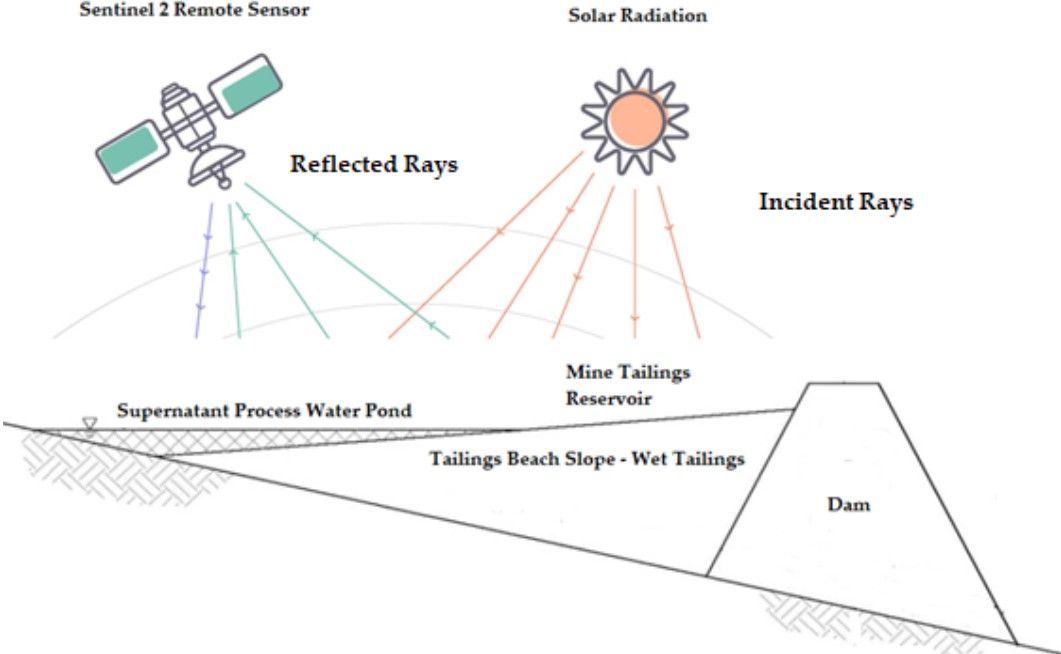

**Figure 2.** Schematical view of remote sensing techniques applied to tailings storage facility monitoring.

Finally, the novelty of this study is that it presents an analysis of spectral indices based on high-resolution satellite images of different real cases of tailings storage facilities, where the highly dynamic and complex spatial and temporal behavior of the supernatant process water pond is compared and interpreted, considering expert field engineering experience, in order to prevent failures and accidents.

## 2. State of the Art of Supernatant Process Water Pond Management in Tailings Storage Facilities

Mine tailings are usually hydraulically handled in pipes or flumes as it is the most economical way to transport them from the concentrator plant to the TSF [32,33]. In opposition, the transport of dry or bulk tailings employing trucks or conveyor belts implies a high investment and operating cost [21]. In general, hydraulically transported tailings are slurry type fluids made up of two phases, the solid (tailing) and the liquid (process water) [21]. The solids concentration of the slurry is a parameter that indicates the proportion of solids and liquid that the mixture has, with a typical concentration of solids by weight of $Cw = 28\%$ obtained in copper mineral concentrator plants. Once the mine tailings leave the flotation cells, they can be taken to thickening equipment, where the use of chemical additives called flocculants accelerates the sedimentation of the tailings, which allows recovering process water and raising the concentration of solids of the mine tailings, reaching the values of $Cw = 50\%$ (thickened tailings) and $Cw = 65\%$ (paste tailings) for the case of copper mining [14].

As previously mentioned, these slurries are transported using pipes (by gravity and by pumping) or flumes (by gravity) where, depending on the solids content by weight of the mixture, the fluid may have a rheology of Newtonian or non-Newtonian behavior. The use of hydraulic pumps is usually required when there is strong uneven terrain to overcome and rheology with non-Newtonian behavior with high levels of viscosity [14,16].

Regardless of the hydraulic transport method considered, mine tailings must be stored in a TSF, a site designed and built for their safe and controlled containment, avoiding leaks and any contact with the environment. Mine tailings usually at the TSF site are transported in pipelines and are discharged from the crest of the dam and the perimeter of the tailings facility at different discharge points, also known as spigots [14].

In this place, a natural phenomenon called sedimentation occurs, which consists of the solid–liquid separation of the deposited tailings. In this way, a supernatant process water pond is formed in the TSF, which must be adequately managed and controlled (Figure 3). It is important to mention that this supernatant process water pond may contain water used to transport the mine tailings as well as rainwater that may fall directly on the TSF. To prevent runoff water from the basin where the TSF lies from coming into contact with the process water from the supernatant pond, the collection works made up of channels are usually built [11].

The waters of the supernatant pond, also known as process waters (waters that have come into contact with mine tailings), usually have a turquoise green color; this is due to the presence of high concentrations of metallic elements (e.g., Cu, Mo, Cd, As, Pb, Mn, and Fe, among others), nonmetallic elements (e.g., sulfates, nitrates, chlorides, and ammonium, among others), and chemical reagents (e.g., copper sulfate, sodium cyanide, sodium ethyl xanthate, pine oil tar, fatty acid soaps, collectors, dithiophosphates, foaming agents, flocculants, and lime, among others) [11].

These process waters must be managed in a controlled manner, avoiding any type of discharge into the environment without prior treatment. It is common practice to monitor the chemical quality of process waters and verify that they comply with the maximum permissible limits of local environmental regulations. Good mining practices consider the recycling and reuse by pumping process water in the metallurgical processes of the concentrator plant (stages of grinding, flotation, and mine tailings transport) and thus avoid the consumption of freshwater of the basin where the mining project lies [11,18,21].

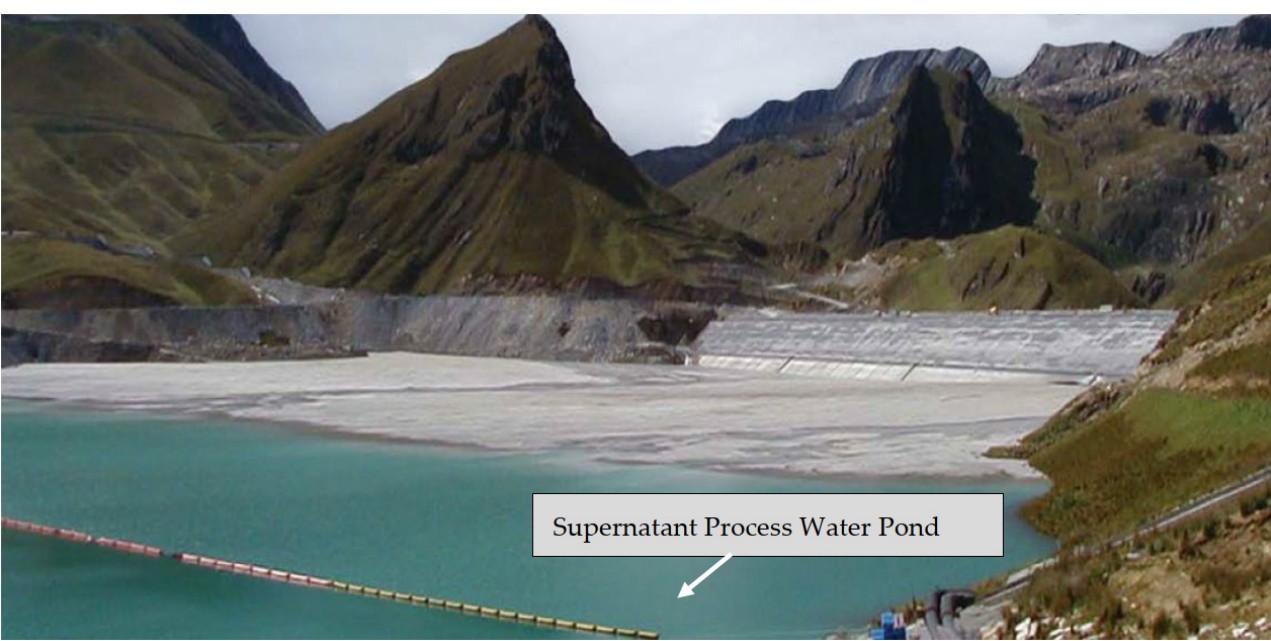

**Figure 3.** Example of panoramic view of tailings dam, tailings beach formation, and supernatant process water pond.

To model the behavior of a TSF, a series of engineering studies are carried out that involve computational modeling, topographic studies, and even laboratory tests to determine the physical–mechanical–chemical parameters of mine tailings. It is important to mention that a TSF is a dynamic infrastructure civil work, which means that it changes over time, due to continuous filling with mine tailings, which implies a growth of the containment dam of the deposit [1,34–36].

Within the mine tailings management design criteria is the definition of the tailings deposit scheme, where the location and a certain number of tailings discharge points, also known as spigots, are assigned. In the area of the discharge point, a fan-shaped tailings beach tends to form, which gives a slope or inclination to the deposited tailings surface (Figure 4). The tailings beach is wet when the tailings discharge point is actively discharging mine tailings, while it is dry when the discharge point is inactive or closed. Supernatant pond water tends to form away from the area of tailings discharge points. The deposition slope of the tailings beach is variable depending on the location closest or farthest from the tailings discharge point. Naturally, due to physical–hydraulic phenomena, segregation of particles corresponding to the solids of the mine tailings is produced, leaving the coarse solids closer to the discharge point and the finer solids further from the discharge point. For conventional tailings disposal (CTD), the tailings beach deposition slope is in the typical range of 0.5% to 1.0%, while for thickened tailings disposal (TTD) the tailings beach deposition slope is in the typical range of 1.0% to 2.0%, and, finally, for paste tailings disposal (PTD), the deposition slope of the tailings beach is in the typical range of 2.0% to 3.0% (Figure 5) [14,37].

Conservative good practice and design implement tailings discharge points from the crest of the dam and the perimeter of the tailings facility to ensure that the location of the supernatant process water pond is as remote as possible from the dam. In this way, the risk of occurrence of liquefaction and tubing phenomena in the TSF dam is reduced [17]. Usually, to control the volume and level of the pond, floating hydraulic pumps are installed and thus the process water is recirculated to the concentrator plant for reuse [18].

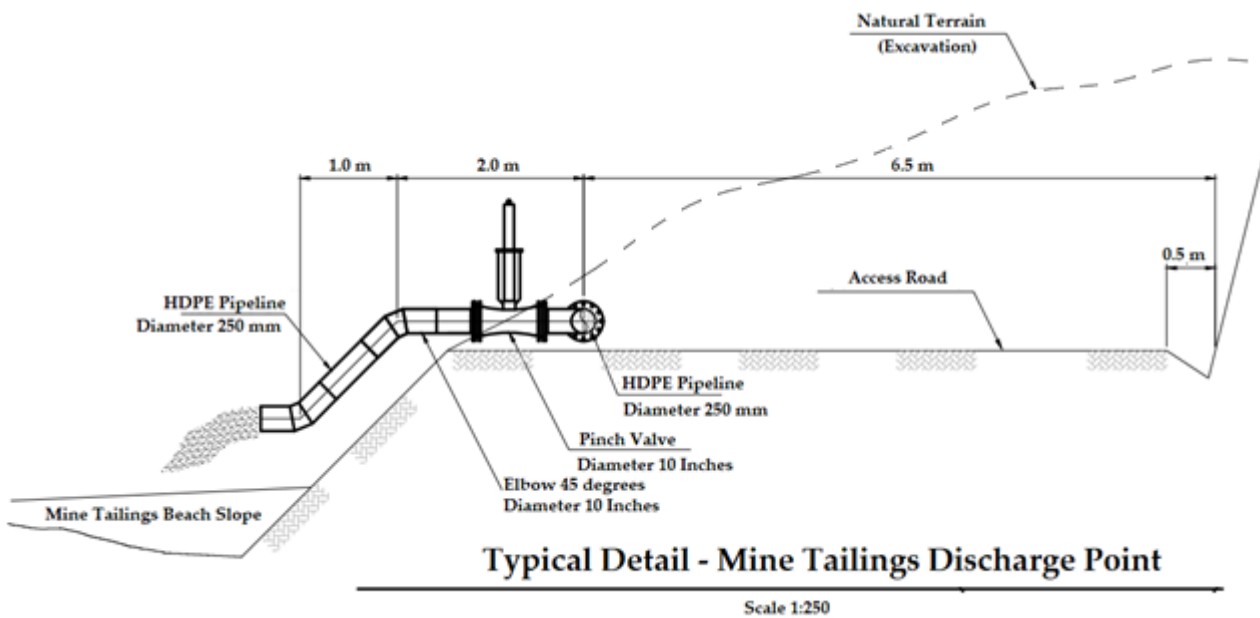

**Figure 4.** Example of Mine Tailings Discharge Point (Spigot).

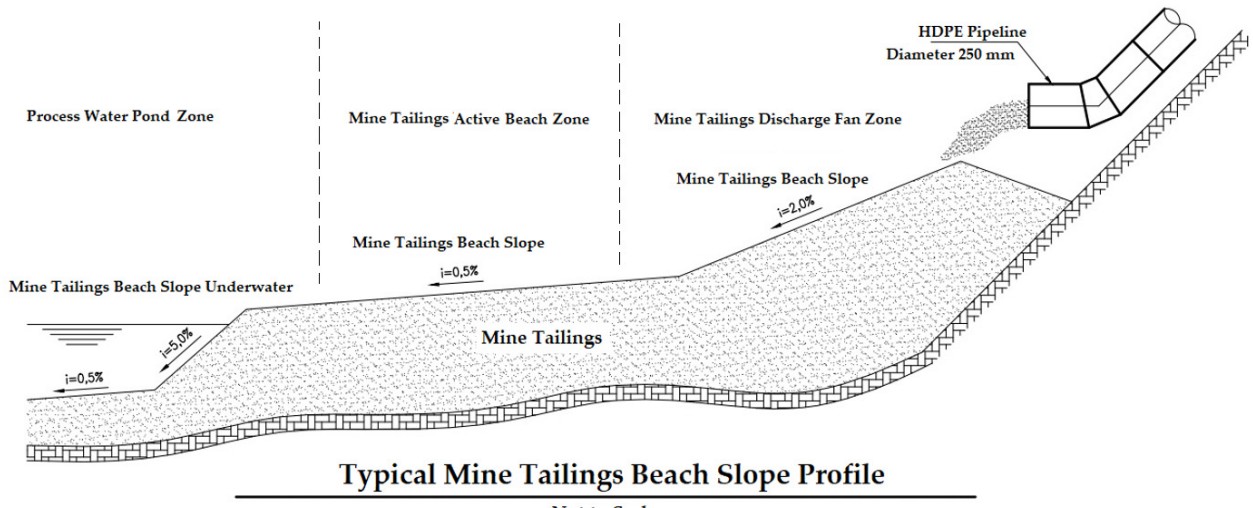

**Figure 5.** Example of tailings beach deposit scheme.

Another important aspect to consider is the management and control of leaks in the tailings deposit. To this end, the following systems are usually implemented: (i) lining with a geotextile layer and HDPE geomembrane, (ii) basal drainage systems, (iii) cut-off trenches, (iv) plastic concrete slurry wall, (v) grout curtain, and (vi) seepage collection reservoirs [11].

Considering the topography of the TSF site, the geometric configuration of the TSF dam, the tailings deposition scheme, and the filling rate of mine tailings, with the use of computational tools, modeling of growth and filling is carried out of the tailings deposit for the different months and years of operation (Figures 6–8). With this information finally presented in engineering plans, it allows for the preparation of the operation manuals and technical specifications for handling the TSF [15].

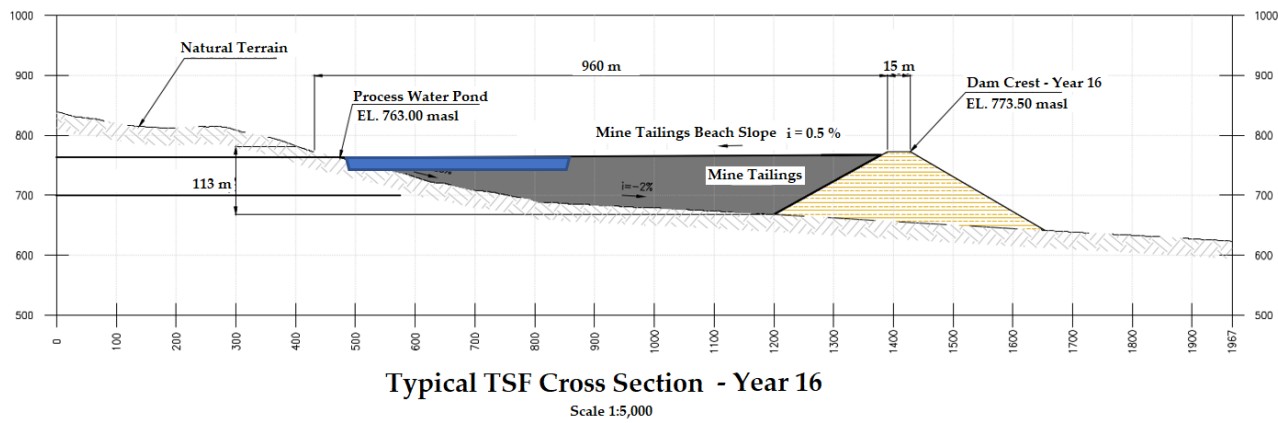

**Figure 6.** Tailings storage facility growth and filling modeling 1 of 3.

**Figure 7.** Tailings storage facility growth and filling modeling 2 of 3.

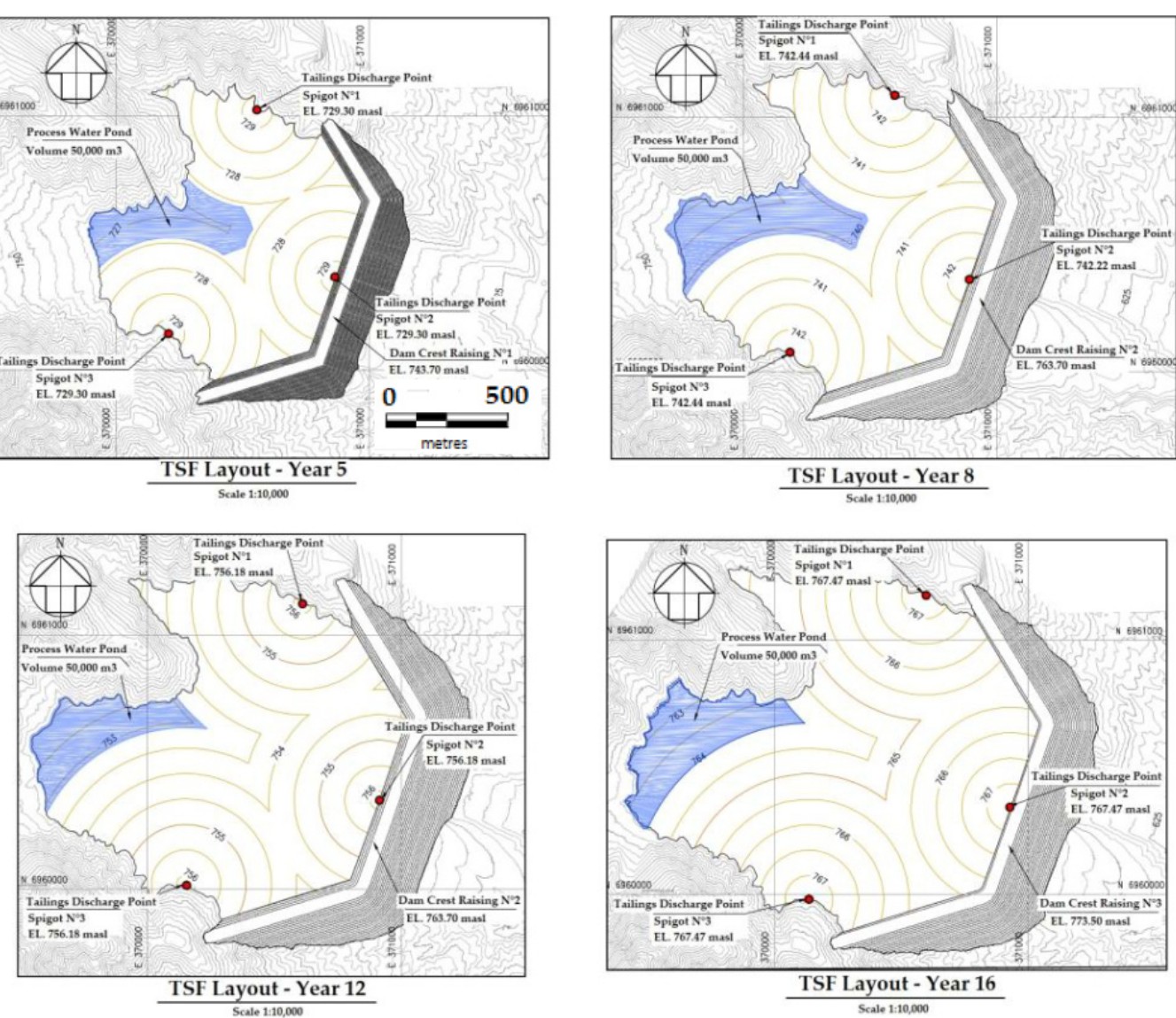

**Figure 8.** Tailings storage facility growth and filling modeling 3 of 3.

With this information, it is important to coordinate between the engineering team and the construction team (contractor) for the construction of the different growth stages of the dam as the deposit is filled with mine tailings. It is key to carry out adequate planning of these tasks and not have problems, providing the minimum freeboard at all times to guarantee the safety of the dam and avoid overtopping events (overflow). The contractor companies that participate in the construction project of the tailings deposit dam must be certified companies with sufficient experience in this type of work; in addition, all quality assurance and control procedures during the construction process must be applied, thus guaranteeing the highest standards [38,39].

Other aspects that should be considered when modeling the filling of the tailings deposit are (i) variability over time of the density of the tailings deposited, for which it is necessary to know the properties of the specific gravity of solids, and the void ratio of the mine tailings, the latter related to the consolidation phenomenon that manifests itself over time; (ii) slope deposition in the tailings beach area, which will depend on the technology of tailings considered [25].

Another issue to consider in modeling the filling of the tailings deposit is rainfall in the sector. Precipitation waters can fall directly on the surface of the tailings deposit, increase the volume of the process water pond, and modify the deposition slopes in the tailings

beaches, reducing the tailings deposition slope values when there is the presence of rain. Additionally in the tailings deposit, depending on the tailings deposit scheme, there can be active or wet tailings beach areas and inactive or dry tailings beach areas [17].

Tailings dam growth and modeling fill engineering design typically involve civil engineers, hydraulic engineers, geotechnical engineers, software modelers, surveyors, CAD draftsmen, and CAD designers. Within the engineering studies, it is necessary to project a growth and filling curve of the TSF, where the elevations of the crest of the dam and the tailings beach are clearly shown to guarantee a minimum freeboard at all times and thus avoid any possibility of overtopping. This information is also necessary to schedule the operation and construction of the different stages of the tailings dam by mine operators and contractors (Figure 9) [14].

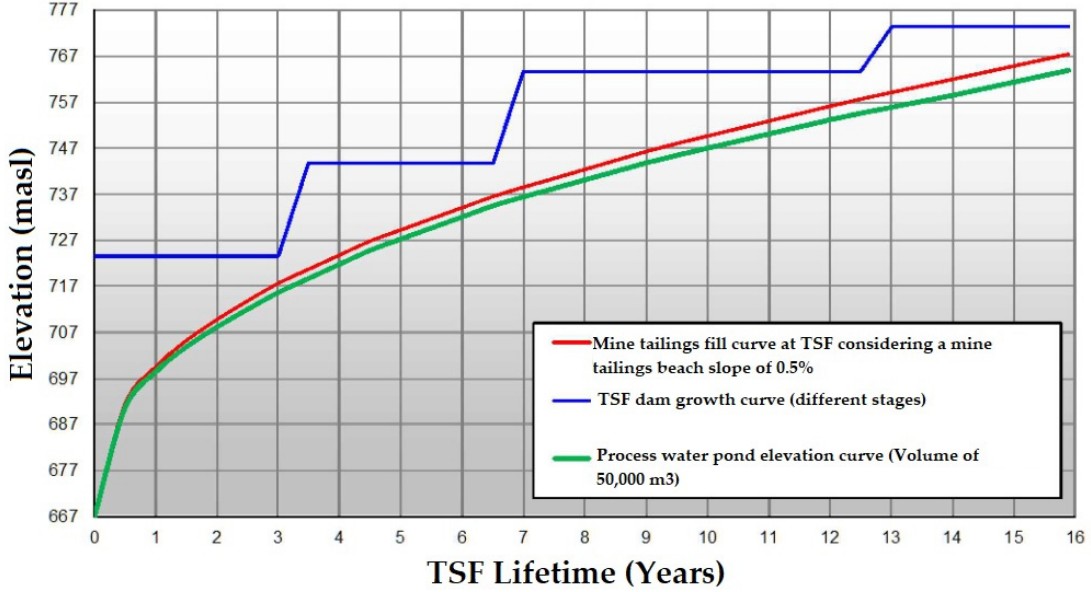

**Figure 9.** Tailings storage facility filling mine tailings curve and dam growth curve.

## 3. Materials and Methods

### 3.1. Characteristics of Tailings Storage Facilities Considered as Study Cases

The TSFs selected for the analysis were chosen due to the regular location of the supernatant process water pond and the availability of very-high-resolution satellite images from the Sentinel 2 sensor through the access offered by the Google Earth Engine (GEE) tool. Additionally, each case study is classified according to its tailings management mode into (i) conservative tailings management, (ii) intermediate tailings management, and (iii) nonconservative tailings management. The main characteristics of the TSF considered as study cases in this research are presented below:

#### 3.1.1. Quebrada Honda Tailings Storage Facility—Peru

Quebrada Honda TSF is located in the department of Tacna in southern Peru, in the middle of the Atacama Desert. It is a deposit that stores copper tailings produced by the Cuajone and Toquepala mines, generated at a production rate of 170,000 mtpd. The tailings dam is constructed with cyclones tailings sands using the centerline construction method (Figure 10). Fine tailings or slimes are discharged from the crest of the tailings dam to form a tailings beach next to the dam and thus keep the supernatant process water pond away from the dam. For this reason, this TSF is classified with conservative tailings management. This TSF has not had a tailings spill, overtopping, or failure such as dam breaks during its lifetime [18].

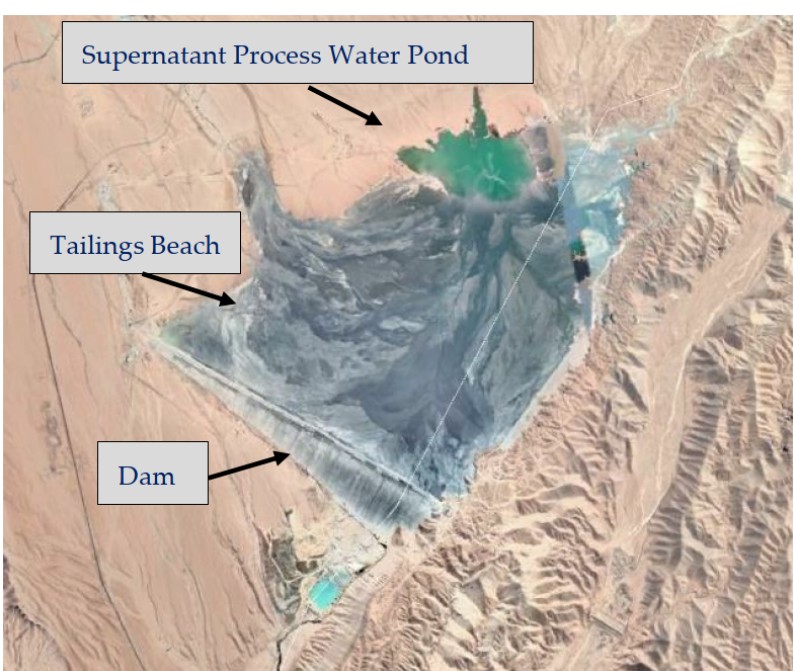

**Figure 10.** Quebrada Honda Tailings Storage Facility—Peru.

3.1.2. Quebrada Enlozada Tailings Storage Facility—Peru

Quebrada Enlozada TSF is located in the department of Arequipa in southern Peru. It is a deposit that stores copper tailings produced by the Cerro Verde mine, generated at a production rate of 240,000 mtpd. The tailings dam is constructed with cyclones tailings sands using the centerline construction method (Figure 11). Fine tailings or slimes are discharged from the crest of the tailings dam to form a tailings beach next to the dam and thus keep the supernatant process water pond away from the dam. For this reason, this TSF is classified with conservative tailings management. This TSF has had no tailings spill, overtopping, or failure such as dam breaks during its lifetime [15,18].

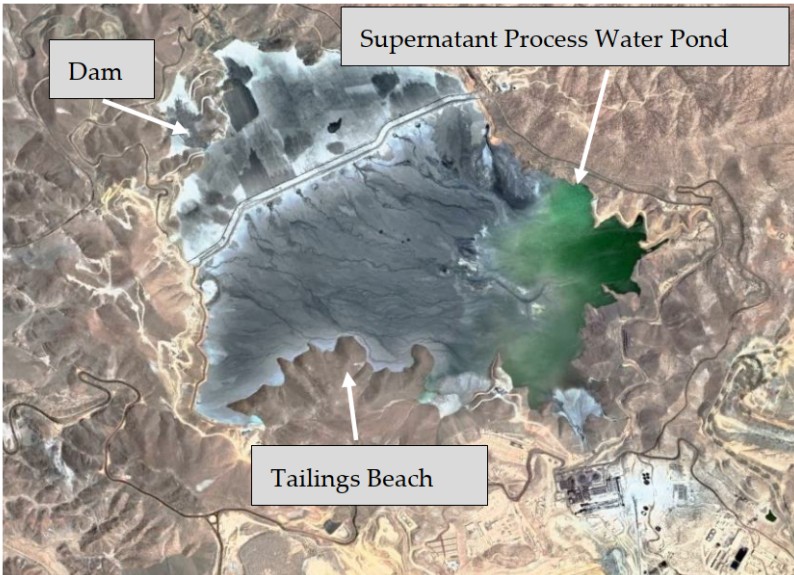

**Figure 11.** Quebrada Enlozada Tailings Storage Facility—Peru.

3.1.3. Laguna Seca Tailings Storage Facility—Chile

Laguna Seca TSF is located in the Antofagasta region in northern Chile. It is a deposit that stores copper tailings produced by the Escondida mine, generated at a production

rate of 400,000 mtpd. The tailings dam is constructed of borrow and rockfill material using the downstream construction method (Figure 12). Tailings are discharged around the entire perimeter of the tailings storage facility, forming a tailings beach that locates the supernatant process water pond next to the tailings dam. For this reason, this TSF is classified with intermediate tailings management. This TSF has not had a tailings spill, overtopping, or failure such as dam breaks during its lifetime [15].

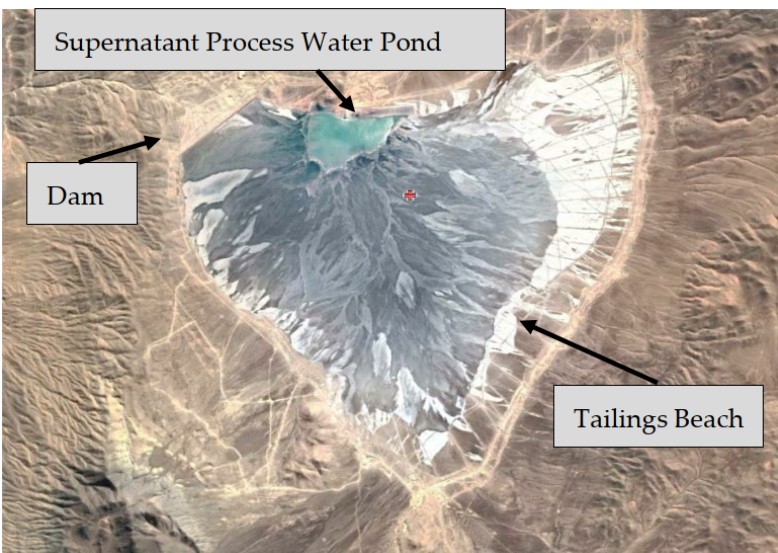

**Figure 12.** Laguna Seca Tailings Storage Facility—Chile.

### 3.1.4. Caren Tailings Storage Facility—Chile

Caren TSF is located in the O'Higgins region in central Chile. It is a deposit that stores copper tailings produced by the El Teniente mine, generated at a production rate of 180,000 mtpd. The tailings dam is constructed of borrow material using the downstream construction method (Figure 13). Tailings are discharged from the rear of the tailings storage facility, forming a tailings beach that locates the supernatant process water pond next to the tailings dam. For this reason, this TSF is classified with intermediate tailings management. This TSF has not had a tailings spill, overtopping, or failure such as dam breaks during its lifetime [15].

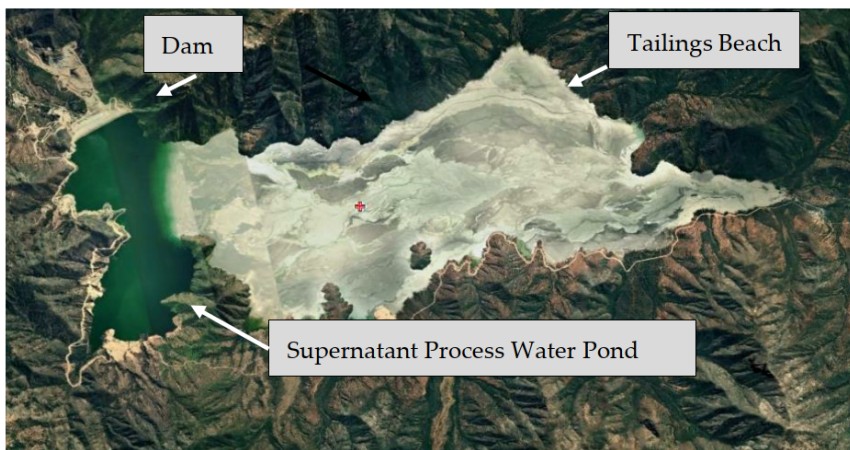

**Figure 13.** Caren Tailings Storage Facility—Chile.

### 3.1.5. Jagersfontain Tailings Storage Facility—South Africa

Jagersfontain TSF is located in the Free State Province in South Africa. It is a deposit that stores diamond tailings produced by the Jagersfontain mine, generated at a production

rate of 20,000 mtpd. The tailings dam is built with borrow material using the centerline construction method around the entire perimeter of the tailings storage facility (ring dyke) (Figure 14). Tailings are discharged from the dam crest around the entire perimeter of the tailings deposit, forming a tailings beach that locates the supernatant process water pond in the center of the deposit [40]. For this reason, this TSF is classified with nonconservative tailings management. This TSF had a failure in its dam on 11 September 2022, causing the spillage of tailings for 8 km, impacting communities and the environment [11].

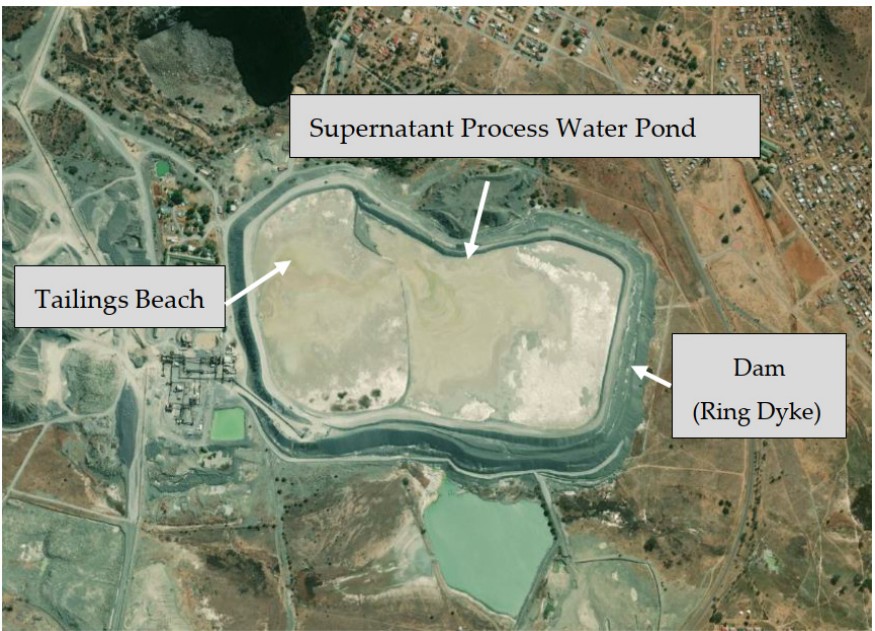

**Figure 14.** Jagersfontain Tailings Storage Facility—South Africa.

### 3.1.6. Williamson Tailings Storage Facility—Tanzania

Williamson TSF is located in the Shinyanga Province of Tanzania. It is a deposit that stores diamond tailings produced by the Williamson mine, generated at a production rate of 15,000 mtpd. The tailings dam is built with borrow material using the centerline construction method around the entire rectangular perimeter of the tailings storage facility (ring dyke) (Figure 15). Tailings are discharged from the crest of the dam around the entire perimeter of the tailings deposit, forming a tailings beach that locates the supernatant process water pond in the center of the deposit. This TSF had a failure in its dam on 7 November 2022, causing the spill of tailings for 6 km, impacting communities and the environment [11].

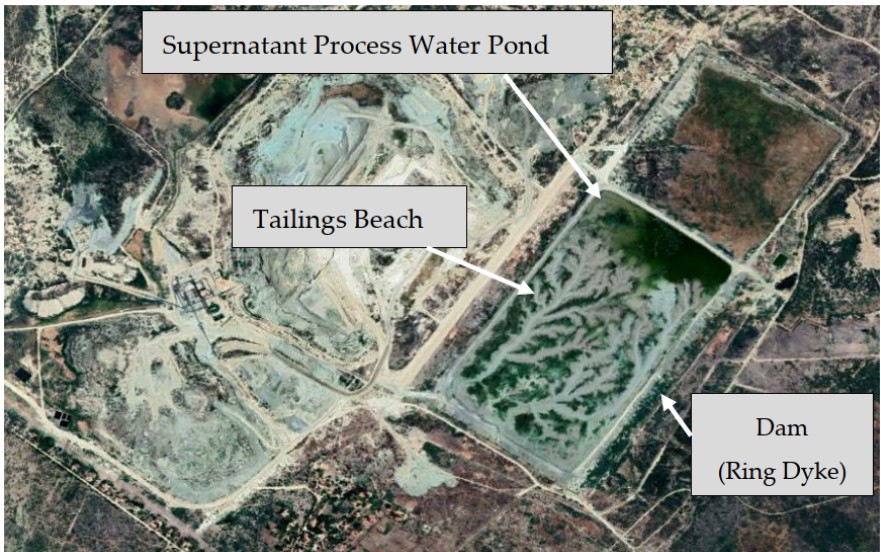

**Figure 15.** Williamson Tailings Storage Facility—Tanzania.

*3.2. Remote Sensing Techniques Applied for Monitoring Tailings Storage Facilities—Analysis and Management of Satellite Imagery*

As the main background for this article, freely accessible satellite images of the Sentinel 2 sensor between 2017 and 2022 were used. From now on, the supernatant process water pond is given the name of supernatant water, and the beach of wet tailings is referred to as wet tailings.

To monitor the presence of supernatant water in the TSF, the NDWI spectral index (Equation (1)) was used, using Sentinel 2 bands B3 (green band) and B8 (infrared band—NIR), taking the value of 0.3 as a threshold from NDWI. The data of monthly raster layers were generated, taking into account the median to ensure the accuracy and certainty of the presence of water at the evaluation site. The NDWI index is calculated per pixel as follows, where green and NIR are the green bands and the near-infrared band of Sentinel 2, respectively:

$$NDWI = \frac{(Green - NIR)}{(Green + NIR)} \tag{1}$$

For the monitoring of wet tailings, the structural characteristic of the material produced was taken into account, which is a residue of the mining–metallurgical process of very fine granulometry, with the presence of chemical substances and a large amount of process water, which make up a kind of mud or paste. Remote sensors such as Sentinel 2 can capture soil moisture in conjunction with the water surface. To capture wet tailings and supernatant water with these characteristics together, a combination of three spectral indices was used: (i) the Modified Normalized Difference Water Index (mNDWI), (ii) Enhanced Vegetation Index (EVI), and (iii) the Normalized Difference Vegetation Index (NDVI). For this, the criteria proposed by Xia et al., where they mention that (mNDWI > NDVI and EVI < 0.1) and (mNDWI > EVI and EVI < 0.1), help to identify the presence of surface water with greater precision. However, it is necessary to mention that this combination of indices was initially created to capture only bodies of surface water. Due to its ability to eliminate interferences (such as wetland vegetation) when considering EVI < 0.1, it increases its ability to capture moisture from the soil because the water signal is stronger than the rest of the materials [41]. The composition of these indices is explained below (Equations (2)–(4)), where, blue, green, red, NIR (near infrared) and Swir1 (shortwave infrared 1) correspond to the surface reflectance of the Sentinel 2 bands (B2, B3, B4, B8, and B11, respectively):

$$mNDWI = \frac{(Green - Swir1)}{(Green + Swir1)} \tag{2}$$

$$NDVI = \frac{(NIR - Red)}{(NIR + Red)} \tag{3}$$

$$EVI = 2.5 \, \frac{(NIR - Red)}{(NIR + 6 * Red - 7.5 * Blue + 1)} \tag{4}$$

The entire process was carried out with the Google Earth Engine (GEE) tool. The method consists of three main steps to be able to access data from areas with the presence of supernatant water and wet tailings (Figure 16).

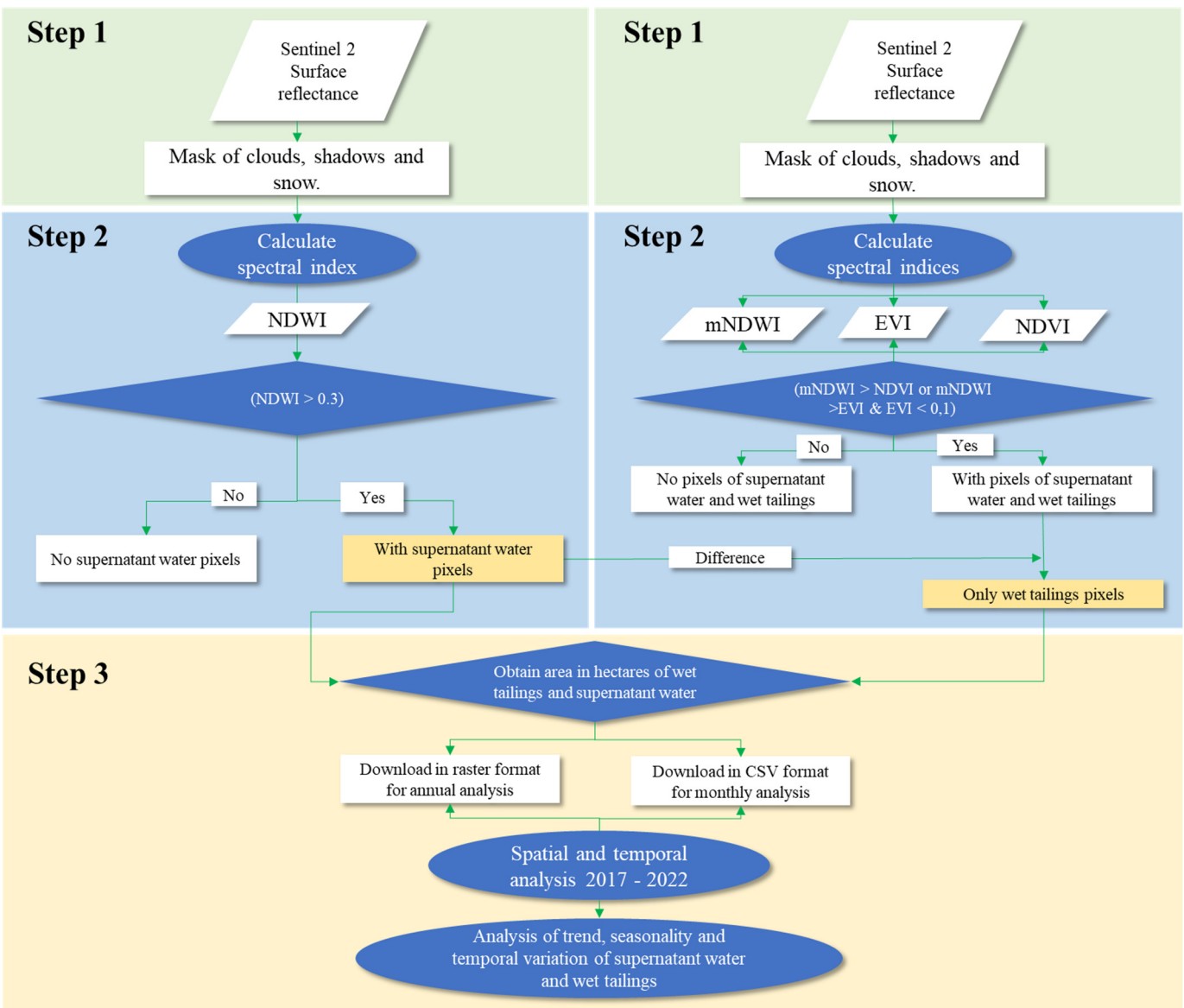

**Figure 16.** Flowchart of satellite image processing to assess supernatant water and wet tailings from TSFs with Sentinel 2 remote sensing. Flowchart adapted from Xia et al. [41].

The first step to check for the presence of supernatant water was to eliminate the pixels with clouds, shadows, or snow from the Sentinel 2 remote sensor by masking clouds, shadows, and snow. In the second step, the NDWI spectral index was calculated, to then apply the criteria mentioned in the second paragraph of this section (origin of Equation (1)) [42–44]. Subsequently, a raster layer with dichotomous pixels was reclassified: pixels = 1 determined supernatant water; pixels = 0 with no supernatant water. In the case of wet tailings, a cloud, shadow, and snow mask tool were also applied as the first step.

In the second step, the mNDWI, EVI, and NDVI spectral indices were calculated, using the criteria described in the third paragraph of this section (origin of Equations (2)–(4)). Subsequently, a raster layer with dichotomous pixels was reclassified: pixels = 1 determined wet tailings and supernatant water; pixels = 0 without wet tailings and supernatant water.

In both cases, to establish the presence of supernatant water and wet tailings, the TSFs were delimited, applying the same principle to the rest of the images within the period 2017–2022 (6 years of study). The data obtained from satellite images generated every 5 days were grouped monthly, that is, between five and six satellite images were grouped to obtain monthly data through the median of the pixels. As a third step, a pixel count was performed for both indices (combined index and NDWI) to determine the total area of supernatant water and wet tailings. To convert from square meters to hectares, the area of each pixel = 0.01 Ha was considered. It must be taken into account that with this last combined index (mNDWI—NDVI—EVI), pixels of supernatant water and wet tailings were obtained. To obtain the definitive wet tailings area, a difference was made with the first index (NDWI) of supernatant water. Both data obtained were downloaded in two formats: annual raster and monthly CSV (comma-separated values) for the spatial and temporal analysis of the TSFs.

### 3.3. Spatial Analysis—Use of Satellite Imagery to Study the Location and Surface Size of Wet Tailings and Supernatant Process Water Ponds in Tailings Storage Facilities

To spatially evaluate the TSFs, the annual dichotomous raster layers of both indices downloaded from the GEE tool were used. For each TSF, the sum of all the rasters was performed using a raster calculator, obtaining pixels with minimum values of 1 (accumulation of material evaluated in one year) and a maximum of 6 (accumulation of material evaluated in 6 years). The purpose was to observe the presence, accumulation, distribution, and location of wet tailings and supernatant water in the TSFs. Considering this as a premise of conservative and safe management of the TSF, the accumulation of supernatant water must be kept sufficiently far from the dam. In addition, it must show a dry or wet tailings beach, with little variability, and respect the capacity of the TSFs, to avoid risks of spillage or overtopping that guarantee the safety of the TSF. The information processing and the generation of maps were carried out in the software of Geographic Information Systems (GIS) of free access QGIS version 3.18.

### 3.4. Temporal Analysis—Use Satellite Imagery to Understand the Dynamic Behavior of Wet Tailings and Supernatant Water Ponds of Tailings Storage Facilities

Time series were used for the temporal analysis of the TSFs, to statistically characterize the temporal behavior of the area covered by wet tailings and supernatant water. The TSFs were classified according to the type of management used: (i) TSF with conservative management, (ii) TSF with intermediate management, and (iii) TSF with nonconservative management. The analysis focused on evaluating three fundamental statistical characteristics in the time series: (i) trend, (ii) seasonality, and (iii) variability. The processing of statistical information is explained below:

- To evaluate the trend and its significance, the Mann–Kendall test was used at a minimum significance level of 95%, as well as the coefficient of determination ($R^2$) and slope analysis.
- For seasonality, the combined seasonality test was carried out, consisting of the Kruskal–Wallis seasonality test, the Friedman seasonality presence test, and the evaluative seasonality test, used in the open-source software specialized in series: JDemetra + 2.2.4 (available at: https://jdemetradocumentation.github.io/JDemetra-documentation/; accessed on 14 December 2022), providing textual seasonality results (present, probably not present, and not present), at the level of 95% confidence (more detail at https://jdemetradocumentation.github.io/JDemetra-documentation/pages/theory/Tests_combined.html; accessed on 14 December 2022).

- To evaluate the variability, the correlation was made between the annual variation coefficient (CV, expressed as a percentage) and the annual average of the area of each material evaluated.

To facilitate the evaluation and reduce the differences in the extreme values of the areas of wet tailings and supernatant water, the data were transformed by logarithm in base 10. All the evaluation analysis was complemented with the use of MS Excel and Statgraphics Software Centurion 2019.

## 4. Results

The main results and findings of the analysis with satellite images and spectral indices of the tailings storage facilities considered as case studies are presented below:

### 4.1. Detection of Presence of Supernatant Process Water Pond and Wet Tailings in TSFs

This study used two metrics to identify the presence of supernatant water and wet tailings in tailings dams. One is the Normalized Difference Water Index (NDWI), used to identify bodies of water and which is a function of measured radiation in the green and NIR regions of the spectrum. The second metric used for wet tailings detection was the combination of three spectral indices: (i) the Modified Normalized Difference Water Index (mNDWI), (ii) the Enhanced Vegetation Index (EVI), and (iii) the Normalized Difference Vegetation (NDVI). Below, Figures 17–22 show the results obtained with these metrics for each tailings deposit analyzed as a case study, showing the accumulated presence during 6 years of operation (period 2017–2022).

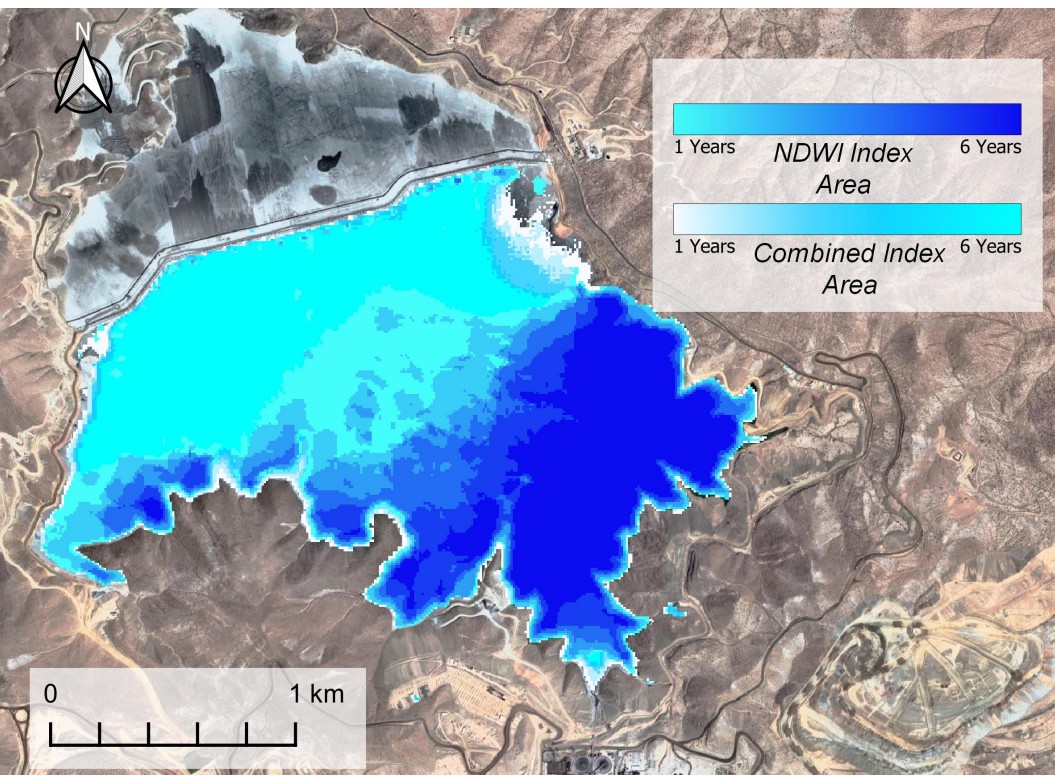

**Figure 17.** Result of applying: NDWI metric and mNDWI—EVI—NDVI combined metric, Quebrada Enlozada TSF case—Peru.

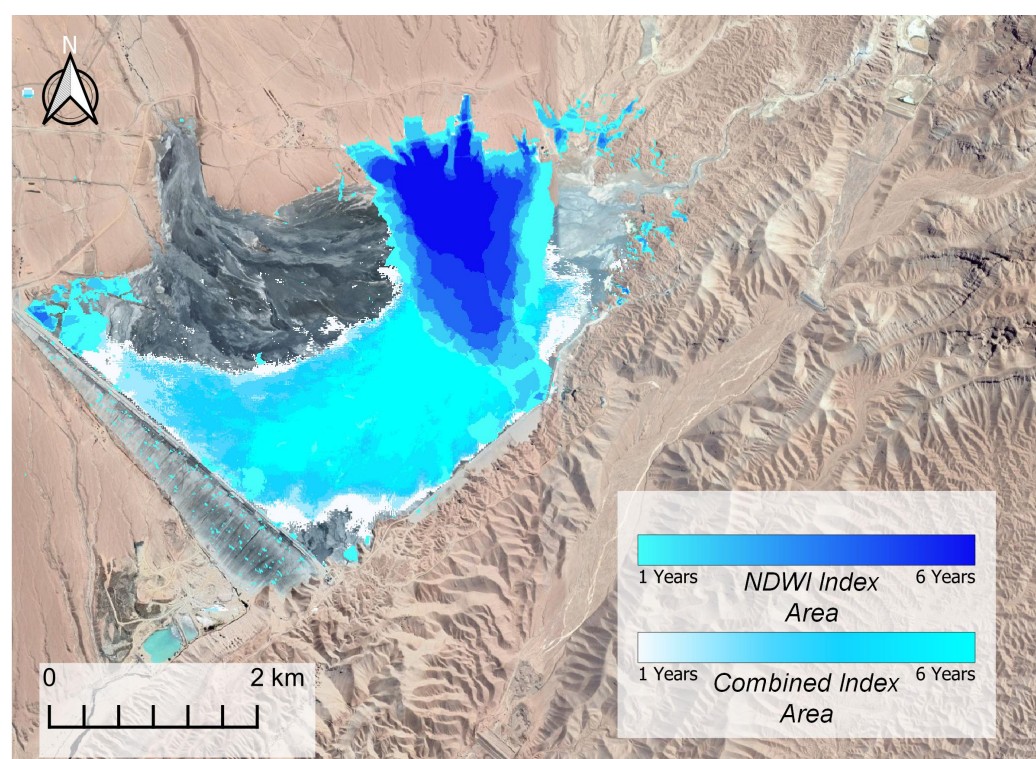

**Figure 18.** Result of applying: NDWI metric and mNDWI—EVI—NDVI combined metric, Quebrada Honda TSF case—Peru.

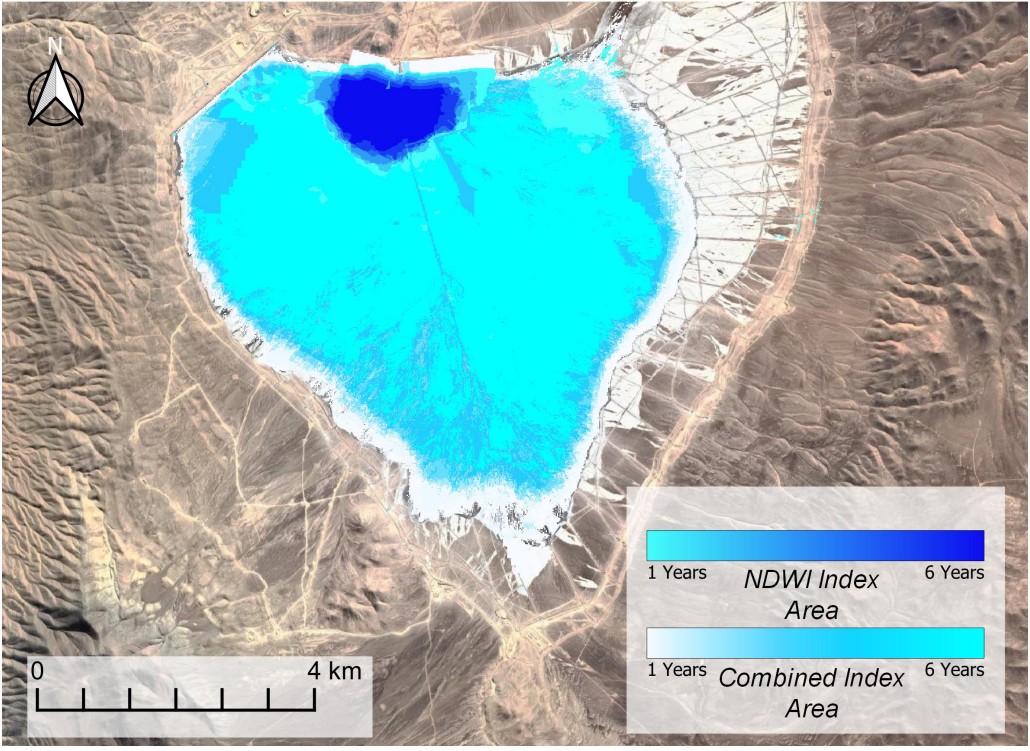

**Figure 19.** Result of applying: NDWI index metric and mNDWI—EVI—NDVI index combined metric, Laguna Seca TSF case—Chile.

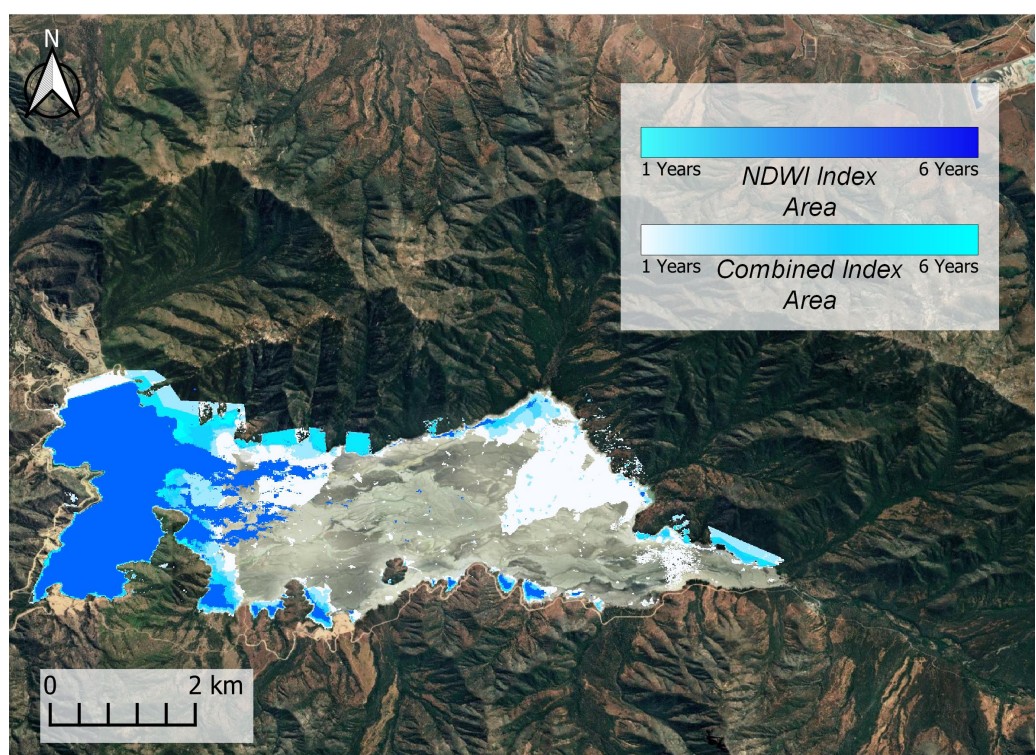

**Figure 20.** Result of applying: NDWI metric and mNDWI—EVI—NDVI combined metric Caren TSF case—Chile.

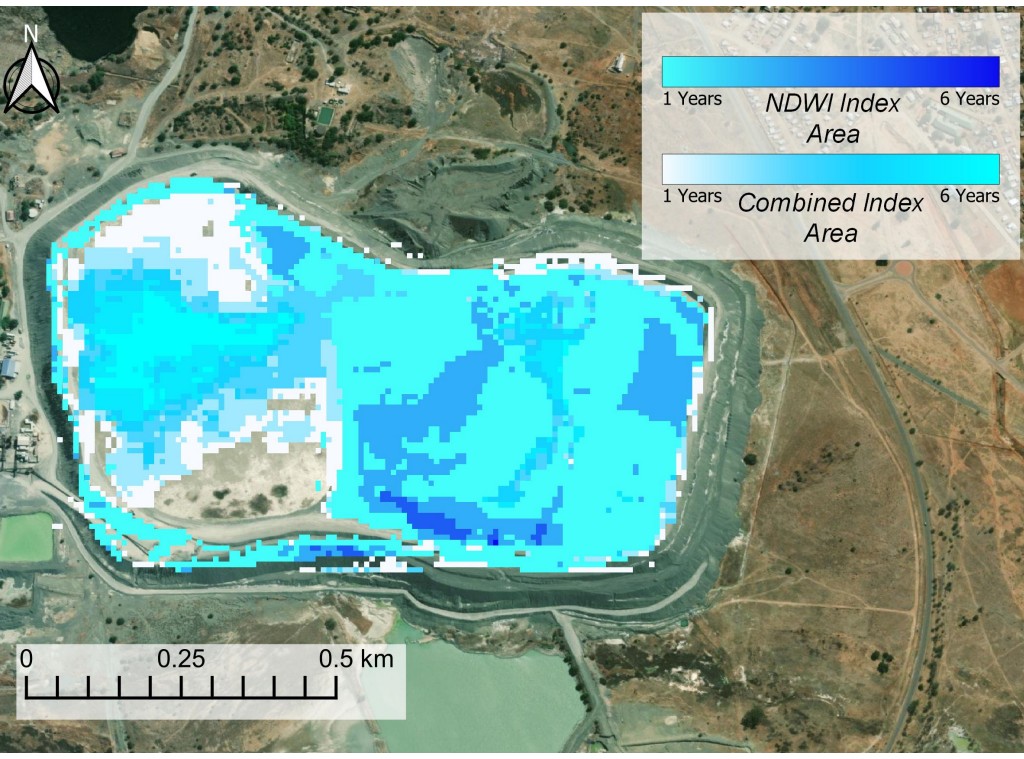

**Figure 21.** Result of applying: NDWI metric and mNDWI—EVI—NDVI combined metric, Jagersfontain TSF case—South Africa.

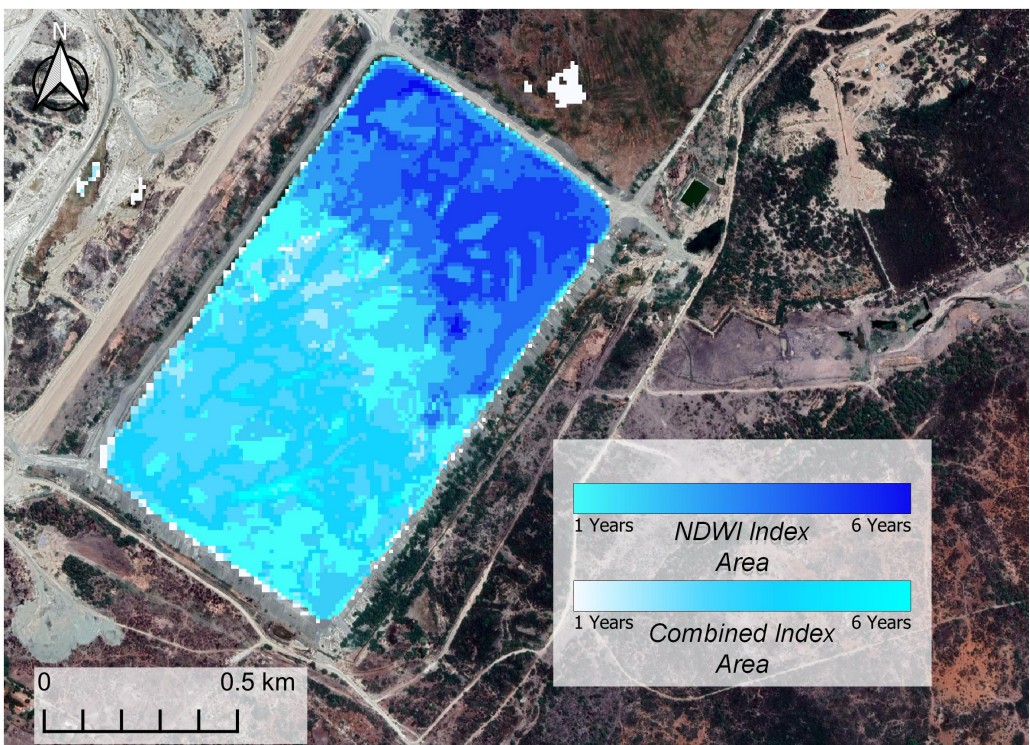

**Figure 22.** Result of applying: NDWI metric and mNDWI—EVI—NDVI combined metric, Williamson TSF case—Tanzania.

It is possible to appreciate in Figures 17–22 the detection of the location and identification of the size of the supernatant water pond in each TSF study case, calculations made based on the values of the NDWI index and combination of mNDWI, EVI, and NDVI indices. In addition, it is also possible to identify the wet tailings zones or tailings beaches for each case. In this way, it is possible to detect if the supernatant process water pond is near or far from the tailings dam.

*4.2. Spatial Analysis of Wet Tailings and Supernatant Process Water Pond in Tailings Storage Facilities*

In this case, the results are focused on analyzing the spatial characteristics of the TSFs evaluated according to the management mode:

- Conservative management of tailings;
- Intermediate management of tailings;
- Nonconservative management of tailings.

The TSFs with conservative management (blue box in Figure 23) present a fairly homogeneous distribution, with increasing tailings accumulation every year. The accumulation of tailings is located in the vicinity of the dam, forming a tailings beach (cases of Quebrada Enlozada TSF and Quebrada Honda TSF). The TFSs with intermediate management (green box), despite maintaining a homogeneous distribution and permanence in their location of both wet tailings and supernatant water, are found in sectors close to the tailings dam, thus increasing the risk of seepage and piping effects through the tailings dam. On the other hand, nonconservatively managed TSFs (red box) show distinct spatial peculiarities. They are characterized by having a heterogeneous distribution of wet tailings and supernatant water. The accumulation of supernatant water is located near the tailings dams; its presence is usually not constant over time and does not show signs of accumulation in several years (for example, in the case of Jagersfontein TSF). Likewise, at one point, the wet tailings came to occupy almost the entire extension of the TSF reservoir, without presenting an accumulation of supernatant water (see Figure 23). It is necessary to mention that the

tailings evaluated with nonconservative management ended in an accident with a failure in its dam, as in the cases of Jagersfontain TSF and Williamson TSF.

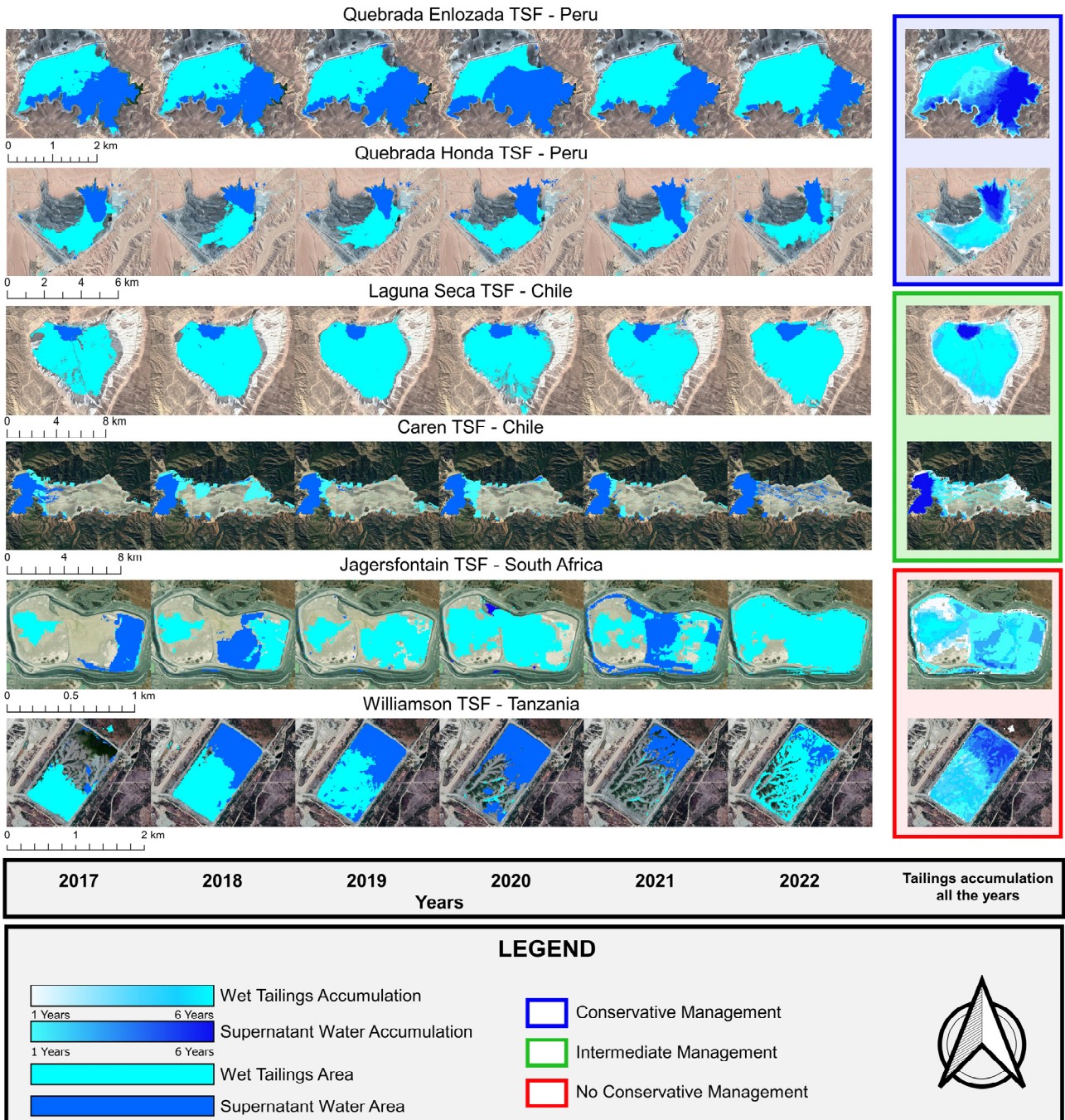

**Figure 23.** Distribution and spatial location of wet tailings and supernatant water from the TSFs evaluated in the period 2017–2022. In the blue, green, and red boxes, the TSFs are separated according to the type of management (conservative, intermediate, and nonconservative). They also show the accumulation of wet tailings during all years, as detailed in the legend.

### 4.3. Temporal Analysis of Wet Tailings and Supernatant Process Water Pond in Tailings Storage Facilities

The temporal behavior is analyzed through time series, both the wet tailings and supernatant water from the TSFs. According to the classification by the type of management, all of them show their characteristics in terms of trends, seasonality, and variability. Figure 24

shows the monthly temporal behavior of the wet tailings area (green line), supernatant water (blue line), and trend lines for each of them. The six TSFs classified according to the type of management were evaluated through Sentinel 2 satellite images during the period 2017–2022. In a complementary way, Table 1 shows the analysis of the significance of the trend and its model lineally generated for the analysis of the graphical slope, the intercept, and the $R^2$, as well as the seasonality test. In general, the wet tailings show a significant positive trend, mainly in the tailings with conservative and intermediate management, while the water supernatant does not show a significant trend.

The TSFs with conservative management were shown to be constant in all months, both in wet tailings and supernatant water. In both cases, they show a significant positive trend in wet tailings according to the Mann–Kendall test (Quebrada Enlozada TSF: $R^2 = 0.27$; slope = 1.0018/month; $p < 0.001$ and Quebrada Honda TSF: $R^2 = 0.06$; slope = 1.8985/month; $p < 0.001$; see Table 1). In both cases, the supernatant water shows a nonsignificant negative and positive trend, respectively ($p > 0.05$), remaining constant over time. The TSFs with intermediate management show a positive trend of wet tailings, significant for Laguna Seca ($p < 0.001$, $R^2 = 0.48$; slope = 31.355/month) and not significant for Caren TSF ($p > 0.05$; $R^2 = 0.04$; slope = 0.0288/month).

**Table 1.** Statistical characterization of the trend and seasonality of the temporal behavior of the wet material and supernatant water in the TSFs evaluated.

| Mine Tailings Management Mode | Tailings Storage Facility Name | Maximum Surface Ha | Tendency ($R^2$) | Tendency Line Model | Tendency (Significance of Mann–Kendall) | Combined Seasonality Test |
|---|---|---|---|---|---|---|
| | Quebrada Enlozada | | | | | |
| Conservative Management | Wet Tailings | 338.67 | 0.2715 | y = 1.0018x + 185.77 | 0.000 ** | Not present |
| | Supernatant Water | 195.17 | 0.0273 | y = −0.2148x + 101.58 | 0.165 | Not present |
| | Quebrada Honda | | | | | |
| | Wet Tailings | 893.88 | 0.0612 | y = 1.8985x + 495.95 | 0.008 ** | Not present |
| | Supernatant Water | 248.16 | 0.0028 | y = 0.1207x + 120.17 | 0.781 | Not present |
| | Laguna Seca | | | | | |
| Intermediate Management | Wet Tailings | 4257.745 | 0.4842 | y = 31.355x + 1562.7 | 0.000 ** | Present |
| | Supernatant Water | 285.16 | 0.0000 | y = 0.0098x + 156.49 | 0.865 | Present |
| | Caren | | | | | |
| | Wet Tailings | 2494.585 | 0.0402 | y = 4.9121x + 310.31 | 0.194 | Probably not present |
| | Supernatant Water | 995.886 | 0.0288 | y = −0.6837x + 550.07 | 0.229 | Present |
| | Jagersfontein | | | | | |
| | Wet Tailings | 63.345 | 0.0093 | y = −0.1356x + 51.751 | 0.008 ** | Present |
| Non Conservative Management | Supernatant Water | 19.753 | 0.0027 | y = −0.0261x + 10.155 | 0.003 ** | Not present |
| | Williamson | | | | | |
| | Wet Tailings | 91.56 | 0.0962 | y = 0.294x + 16.073 | 0.968 | Not present |
| | Supernatant Water | 45.14 | 0.0170 | y = −0.0186x + 1.5148 | 0.761 | Not present |

** correspond to the Mann–Kendall level of significance at 1%.

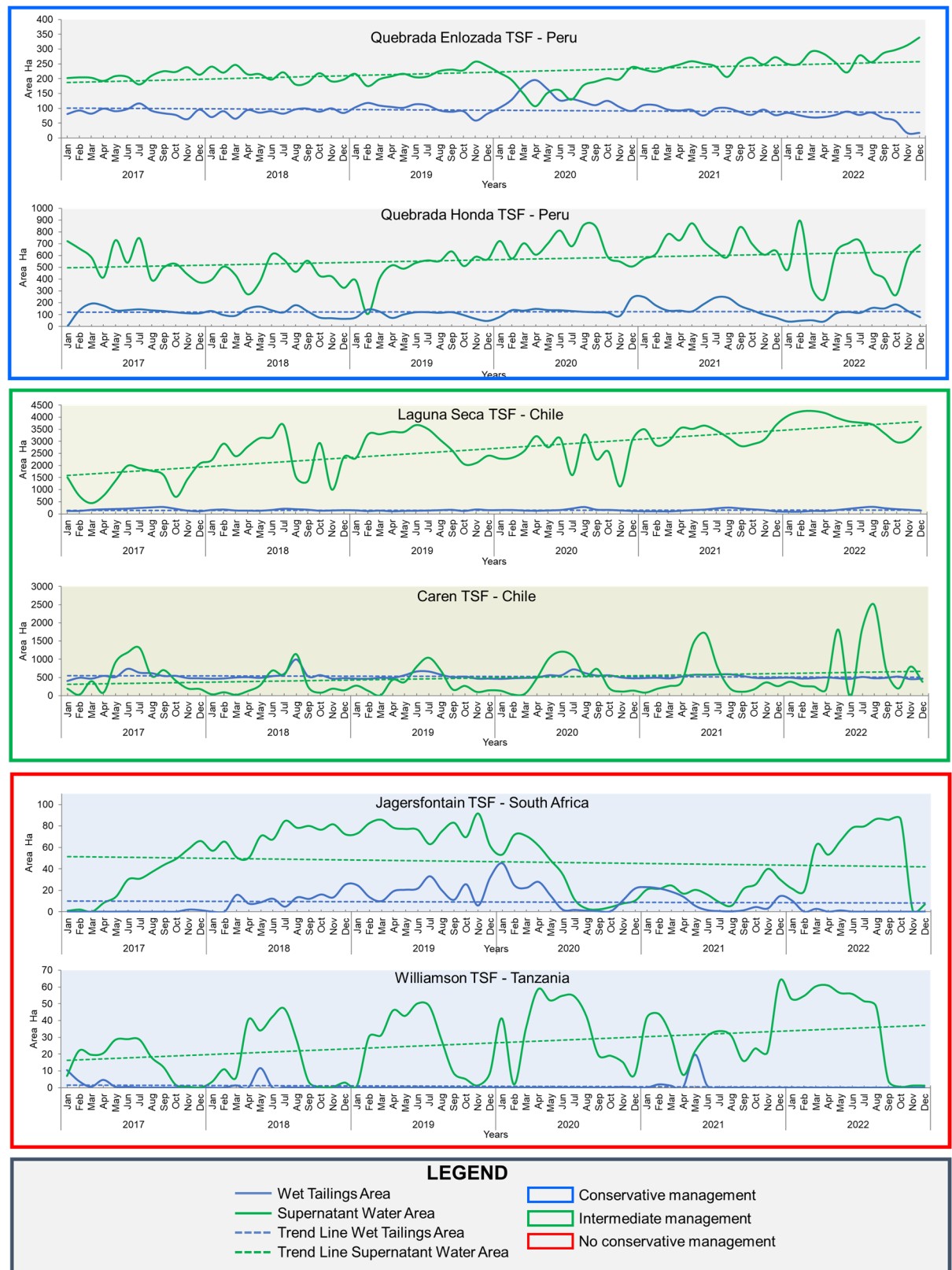

**Figure 24.** Time series of wet tailings areas and supernatant water from the evaluated TSFs. The lines between green and blue crosses on the timelines correspond to the trend of the wet tailings and water supernatant areas, respectively. The boxes with blue, green, and red lines correspond to the types of management of the TSFs (conservative, intermediate, and nonconservative management, respectively).

It should be noted that the case of Laguna Seca TSF is the one with the greatest increase and is the one that shows the largest area of wet tailings in the 6 years evaluated. In both cases, the supernatant water shows a positive and negative trend, respectively, but without significance ($p > 0.05$); that is, they remain constant over time.

In this evaluation, the TSFs that suffered dam collapse accidents causing spillage Jagersfontain TSF from South Africa and Williamson TSF from Tanzania are considered nonconservatively managed (red box; Figures 23 and 24). These show unconventional and unusual statistical characteristics concerning the rest of the TSFs evaluated. For example, Jagersfontein TSF shows a significant negative trend ($p < 0.001$), with $R^2 = 0.0093$ for wet tailings and $R^2 = 0.0027$ for supernatant water. These are the lowest values among the trends with significance. In addition, it is the only one that shows signs of reduction with a significant trend in supernatant water at a rate of $-0.0261$/month. In the case of Williamson TSF, it shows a positive trend for wet tailings and a negative trend for supernatant water, being nonsignificant in both cases.

### 4.3.1. Seasonality Analysis of Wet Tailings and Process Water Ponds in Tailings Storage Facilities

Regarding seasonality, it can be mentioned that the results obtained were indistinct according to the type of management in each TSF. According to the combined seasonality test, both the wet tailings, and the supernatant water, the TSFs with conservative management did not show seasonality. Unlike most of those that have intermediate management, they do present seasonality in both materials evaluated, except the Caren TSF wet tailings, which probably do not present seasonality. On the other hand, in the TSFs that have nonconservative management, the results are indistinct. The Jagersfontein TSF wet tailings exhibit seasonality, while the supernatant water does not exhibit seasonality, which is unusual considering the other TSFs, while in Williamson TSF, there is no seasonality in wet tailings and supernatant water (see Table 1).

### 4.3.2. Variability Analysis of Wet Tailings and Supernatant Process Water Pond in Tailings Storage Facilities

The evaluation of the variability was carried out using correlation graphs between the annual coefficient of variation (CV%) of the area and the annual average of the area for both wet tailings and supernatant water, transformed by logarithm in base 10. These were divided into two graphs, one of the wet tailings and one of the supernatant water.

In general, the variability of wet tailings can distinguish the TFSs according to the type of management used. These are very different and show particularities depending on their size (Figure 24). Those managed conservatively are enclosed in the blue line ellipse. Their location suggests that they are medium-sized TSFs with little variability, that is, the wet tailings are constant over time. The TSFs with intermediate management are the largest (green ellipse) and present little variability, remaining below 10% CV, except Caren TSF in 2022, the year in which its wet tailings had a considerable variation. In the case of TSFs with nonconservative management, they present a lot of variabilities, showing extremes ranging from 10–90% of the CV. These cases correspond to the years 2017–2018 at Jagersfontein TSF and 2018–2019 at Williamson TSF. The distribution contrasts with a power-type regression model, with a high value of the coefficient of determination ($R^2 = 0.87$; CV% $= 53.086 *$ area $-2.808$; see Figure 25).

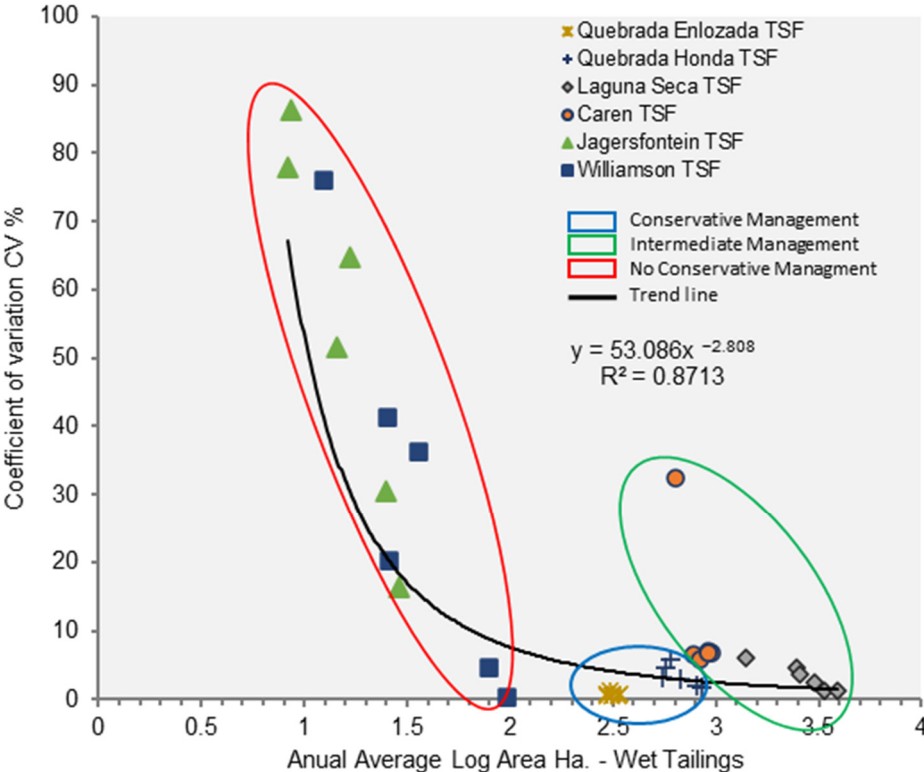

**Figure 25.** Correlation between the coefficient of variation (CV) and the annual average of the wet tailings area. The latter is expressed in hectares with logarithmic transformation. All the TSFs evaluated are separated by ellipses according to the classification of types of management used.

Figure 26 details the variability of the supernatant water of the TSFs. These show a similar distribution to the variability of wet tailings, indicating little variability in larger TSFs, corresponding to those managed conservatively and intermediate (blue and green ellipse, respectively). These do not exceed 30% variability according to the percentage of CV. On the other hand, TSFs with nonconservative management present high variability, exceeding 100% of the CV. This indicates that the standard deviation is greater than the mean, due to the presence of extreme data observed in the year 2022, both for Jagersfontein TSF and Williamson TSF, the year in which the dam breach and tailings spill accident occurred in both cases. The variability distribution presented generated a logarithmic regression model, with $R^2 = 0.88$ and a CV% model = $-41.04 * \ln(\text{area}) + 118.23$ (see Figure 26). All these variabilities analyze using the CV coefficient of variation contrast and complement each other with the time series in Figure 24.

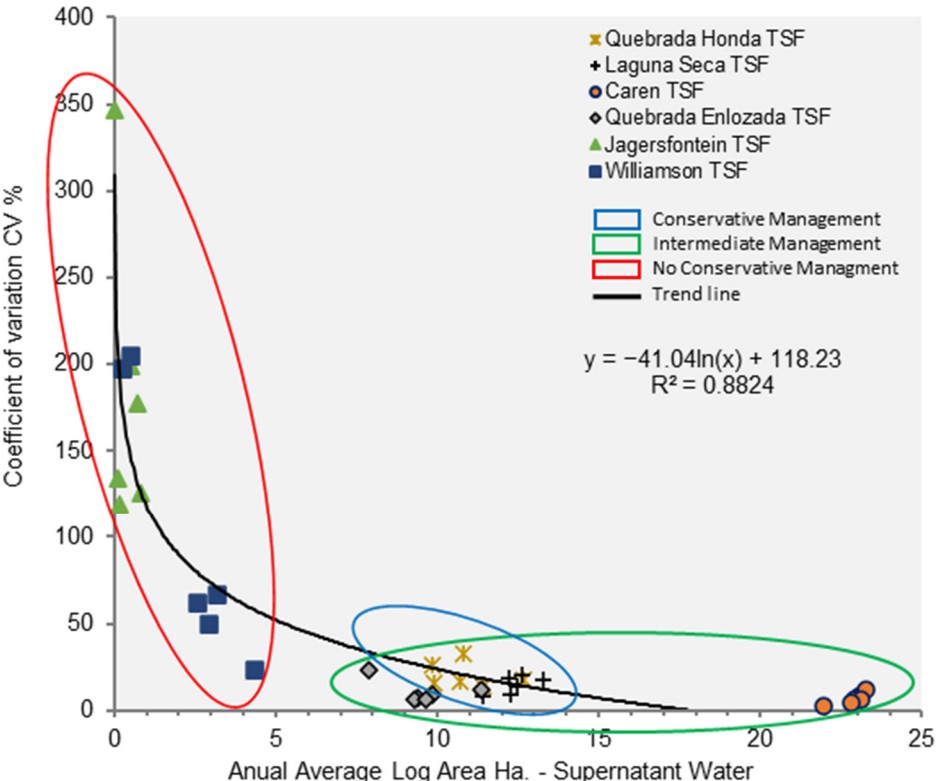

**Figure 26.** Correlation between the coefficient of variation (CV %) and the annual average of the supernatant water area. The latter is expressed in hectares with logarithmic transformation. All the evaluated TSFs are separated by ellipses according to the classification of types of management used.

## 5. Discussion

Remote sensing is a very useful tool for monitoring mine tailings, due to its efficiency, data accessibility, and low costs. This article provides precedents for the evaluation and monitoring of TSFs, with spatial and temporal details that can help determine the level of risk [45]. In addition, the use of satellite images from Sentinel 2 and the implementation of the Google Earth Engine (GEE) tool for the processing and multispectral interpretation of satellite images turns out to be a very powerful and precise instrument to carry out spatial and temporal analysis of systems with highly dynamic and complex behavior such as TSFs [46–48].

It is possible to detect the location and identify the size of the supernatant process water pond in each TSF case study through the calculations made in the Google Earth Engine (GEE) tool based on the values of the NDWI index, and a combination of mNDWI, EVI, and NDVI. In addition, with the use of these indices, it is also possible to identify the wet tailings zones or tailings beaches for each case. In this way, it is possible to detect if there is a low, medium, or high level of risk of failure, taking into account whether the supernatant process water pond is near or far from the tailings dam.

The methodological principle of the use of remote sensing applied in this article is similar to other studies that also monitor wet tailings and supernatant process water ponds with the use of satellite images [49–52]; all of them have in common the use of spectral indices such as NDWI, NDVI, EVI, and mNDWI, independently or together, complemented by the use of bands such as RGB, NIR, and SWIR. These use indistinct methodologies, from basic correlation approaches for the model generation to the use of artificial intelligence and machine learning. The separation of covers carried out by these indices mainly obeys the physical characteristics of the tailings such as particle size and humidity. The greater the area, the greater the absorption of radiation; therefore, it reflects less, and the opposite occurs with fine particles. Likewise, the reflectivity of dry tailings is greater than that of

wet tailings; this difference is appreciated in several bands such as NIR and SWIR when evaluating the spectral signatures of tailings [44]. What stands out in this research is the use of simple and basic spectral indices based on the research of O'Donovan et al. [42] for supernatant water and Xia et al. [41] for wet rewash validated in the field. Another advantage is that the indices used do not discriminate based on the type of mineral that is being exploited, but rather on the structure and physical condition, in this case, the presence of moisture and its reflectance characteristics. This helped the monitoring of TSFs in the period 2017–2022, considering the time series of the areas of wet tailings and supernatant water analyzed. In addition, it was possible to take advantage of the spatial and temporal resolution of the Sentinel 2 images (10 m pixel size every 5 days) from the year 2017 (2015–2016 every 10 days), allowing access to a greater amount of data and eliminating possible monitoring errors. This same characteristic offers outstanding advantages concerning the rest of the remote sensors such as Landsat 8 for example (temporal resolution 16 days and pixel size 30 m).

Both the velocity of the mine tailings from the discharge pipe (spigot) and the difference in elevation between the point of discharge and the tailings beach influence the length and width of the tailings beach, so it is necessary to consider these factors, aspects in the daily operation of the tailings deposit. Thus, the training of mine operators is important, having experienced knowledge in the use of discharge valves, activation/deactivation of discharge points, topographic controls, and bathymetry in the tailings deposit, among others. It is recommended that the pipe velocities be in a range between 1.5 m/s and 4 m/s to avoid sedimentation and accelerated wear of the pipe, respectively. Finally, the difference in elevation between the tailings discharge point (spigot) and the tailings beach will be defined by the freeboard, where a minimum value of 5 m will be predominant, as a safety measure to avoid overtopping [53].

The particle size of the mine tailings is a key parameter for the slope of the tailings beach and the angle of repose, so it is necessary to carry out studies to determine the relationship between the grain size and the rate of variation of the mine tailings beach. To this end, it is possible to develop mine tailings deposition slope tests on a pilot scale, controlling the parameters of grain size, discharge flow, solids concentration by weight, and tailings rheology. It is important that mining companies become aware that it is necessary to develop these tests constantly, in order to better control the extension of the tailings beach and the location of the supernatant process water pond, in mining–metallurgical production processes under dynamic and complex conditions [54].

The results of the spatial analysis of the TSFs make it possible to clearly identify the area with wet tailings or tailings beach, and also the area of bodies of water or supernatant process water ponds, for each of the cases studied. It is also possible to determine the location and area of each of the supernatant process water ponds at a spatial and temporal level. In this way, the determination of the type of tailings management carried out in each case, whether conservative, intermediate, or nonconservative, is evident.

In the spatial aspect, the location and level of the supernatant process water pond qualitatively indicate the degree of risk of the TSF. It is a very important element that must be constantly monitored from the planning stage of the TSF, considering remote sensors, complemented with field water level sensors [55]. Likewise, the permanence of the supernatant process water pond in the planned space of the TSF helps to monitor possible seepage and percolation of process water that can harm the physical stability of the tailings dam. Another influential factor to highlight is the harsh climate with seasonal rains, in addition to factors such as tailings granulometry, tailings beach slope, surface runoff, and groundwater flow. Along the same lines, the spatial monitoring of wet tailings is of special consideration because in this way, the formation of a tailings beach next to the TSF dam is controlled, moving away the supernatant process water pond and reducing the risks of failure of the dam [56].

Regarding temporal analysis, TSFs allow for defining the trend, seasonality, and variability of the areas of wet tailings and supernatant process water ponds. The time series

obtained for each case allows us to understand the complexity of the dynamic behavior that tailings deposits have during their operation each year through their mouths. Changes in the location and size of the process water supernatant pond are a reality for mine tailings management, a reality that must be understood by TSF operators. Some changes can be explained by the operational reasons of the concentrator plant that generates the tailings, or they can also be influenced by changes in climate associated with the phenomenon of climate change, registering alterations in rainfall patterns (magnitude, intensity, duration, and recurrence).

The tendency of the supernatant process water pond is constant in time and depends on the filtering capacity of the granulometry of the wet tailings, the slope of deposition of the tailings, temperature, and evaporation. Likewise, seasonality is important, especially in places where the climate is variable in seasons according to the time of year. Many times, the management of the supernatant process water pond and the wet tailings will depend on the presence of precipitation, constituting a cycle of drying and wetting (for example, the Caren TSF case), affecting the mechanical and resistant properties of the tailings that can end in an accident. In these cases, excess water to a certain extent can be considered normal if the same regime is maintained, through water level management. These depend on the spatial scale, showing differences in management according to the size of the TSF. The high variability in small spaces is due to the lack of regularity in the depositing and filling processes of the TSF. It is partly acceptable that it presents variability, but if the changes are abrupt, this is a symptom of inadequate management of the TSF . According to the results presented in this article, it is possible to point out that one of the causes of dam failures in the Jagersfontain TSF in South Africa (Figure 27) and Williamson TSF in Tanzania (Figure 28) was due to inadequate and nonconservative management of the supernatant water pond, having areas of the reservoir with water in contact with the dam. This could have produced phenomena such as the liquefaction and piping of the dam materials, causing their collapse [57].

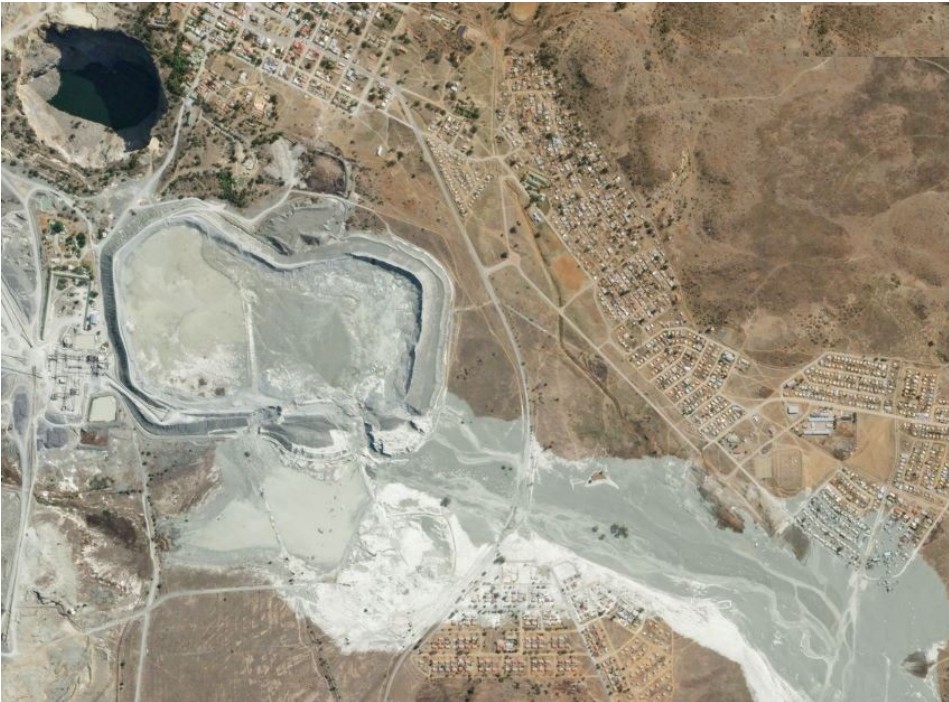

**Figure 27.** Jagersfontain tailings storage facility dam failure, 11 September 2022, South Africa.

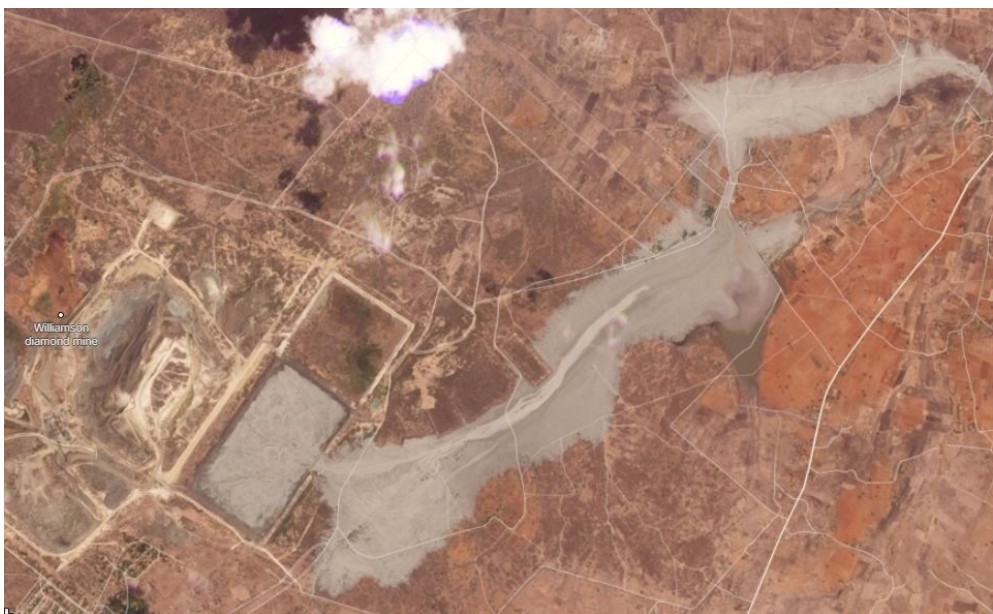

**Figure 28.** Williamson tailings storage facility dam failure, 7 November 2022, Tanzania.

Therefore, based on the spatial and temporal characteristics of the TSFs, is possible to mention that conservative management is characterized by the following: (i) it keeps the supernatant process water pond away from the tailings dam; (ii) the supernatant process water pond is relatively constant in time; (iii) the wet tailings show a significant increase in area; and (iv) the distribution of wet tailings is uniform throughout the entire TSF. Meanwhile, the moderately managed TSFs present the following characteristics: (i) too much seasonality; (ii) the supernatant process water pond is on or near the dam; and (iii) they are fairly variable over time. Finally, TSFs with nonconservative management are characterized by the following: (i) the supernatant process water pond is located next to the dam; (ii) they present a lot of variability; (iii) the supernatant water does not maintain a defined location, many times it disappears with unidentified leaks; and (iv) the distribution of wet tailings is not uniform. Taking the characteristics of these three types of management into account, the level of risk associated with dam failure can be determined.

It is recommended to carry out studies similar to the one presented in this article, but considering a greater number of TSF cases, with different topographical, climatic, and mineralogical characteristics, and thus form a global database. In addition, the aspects evaluated in this article could be integrated as part of the continuous monitoring of TSFs using remote sensing. Experiences which integrate sensors for an emergency monitoring system and assess dam deformation and water contamination could serve as a reference to be incorporated into mine tailings governance programs [58–60]. This can help verify dam deformation through spatial analysis of the water body and cumulative tailings distribution. Likewise, this article can be complemented with the help of specific sensors to evaluate ground movement such as radar sensors (for example, Sentinel 1, which showed good results in different investigations) [61–65]. Finally, it is possible to mention that there is the possibility of exploring the use of satellite images from other freely accessible remote sensors such as Landsat 5, Landsat 7, and Landsat 8, which, despite their spatial resolutions being smaller than that of Sentinel 2, have a large temporal resolution database, which can be complemented in cases where historical data are required.

## 6. Conclusions

According to the different analyses carried out in this article, it was identified that one of the most critical aspects in the management of TSFs is the control of the process water stored inside the TSF reservoir area, whether it is process water from the sedimentation of the tailings (supernatant process water pond) or water from precipitation. Along with

this, the complexity of the interaction with inter- and transdisciplinary coordination of professionals (e.g., engineering companies, construction companies, mining companies, and local government auditors, among others) in the operation of the tailings deposit over time (years) was identified as a potential cause of errors or nonconformities in the performance of good tailings management practices.

The human factor was also identified as a key aspect in tailings deposit failures considering manual monitoring, revealing the lack of real-time intelligent monitoring systems of key critical parameters to detect anomalies in the behavior of physical stability of the tailings deposits. Today it is a necessity for the mining industry to be able to instrument the entire tailings deposit with different types of sensors, and thus have full real-time control of its behavior 24 h a day, 365 days a year.

The analysis carried out in this article confirms that conservative tailings management practices, that is, discharging mine tailings from the crest of the dams to form a tailings beach next to the dam and maintaining keeping the supernatant process water pond as far away as possible, provide security in physical stability and considerably reduce the risks of dam failure.

Thus, for example, in some countries such as Chile, Peru, and Brazil, the construction of tailings dams has been prohibited using the construction method in an upstream direction, and the prohibition of having a supernatant process water pond next to the dam should also be promoted. This would considerably reduce the levels of risk of dam failure during the operation of the TSFs.

Considering that the management of the level of risk of failure in the TSFs is highly variable, dynamic, and complex when the tailings deposited have high amounts of water, as is the case of the implementation of conventional tailings technology (CTD), for example, it is also necessary to promote the massive and aggressive use and implementation of tailings dewatering technologies, such as thickened tailings (TTD), paste tailings (PTD), and filtered tailings (FTD), which allow mining tailings to be stored with small amounts of water, even in some cases without forming supernatant process water ponds.

The use of technologies for monitoring the behavior of TSFs in real time with the help of remote sensors, such as using satellite images, and their processing with tools such as Google Earth Engine (GEE), is nowadays a priority for the mining industry in the Industry 4.0 era. The information provided by these technological tools allows us to make responsible and intelligent decisions in real time, generating calls for attention, alerts, and warnings regarding how the management of mine tailings in the deposit is being carried out.

Finally, this article calls attention to the mining industry, in general, to promote the monitoring of tailings deposits responsibly and transparently with the communities, encouraging capital investment in the implementation of sophisticated and modern systems in the activities of monitoring the behavior of the tailings storage facilities, and thus guaranteeing the safety of communities and the environment.

**Author Contributions:** Conceptualization, C.C. and D.C.; formal analysis, C.C. and D.C.; investigation, C.C. and D.C.; resources, D.C.; writing—original draft preparation, C.C. and D.C. writing—review and editing, C.C. and D.C.; visualization, C.C. and D.C.; supervision, D.C. All authors have read and agreed to the published version of the manuscript.

**Funding:** The research is funded by the Research Department of the Catholic University of Temuco, Chile.

**Data Availability Statement:** The data presented in this study are available on request from the corresponding author.

**Conflicts of Interest:** The authors declare no conflict of interest.



## Abbreviations

| | |
|---|---|
| TSF | Tailings storage facility |
| NDWI | Normalized Difference Water Index |
| mNDWI | Modified Normalized Difference Water Index |
| EVI | Enhanced Vegetation Index |
| NDVI | Normalized Difference Vegetation Index |
| Swir1 | Shortwave infrared 1 |
| NIR | Near infrared |
| GEE | Google Earth Engine |
| GIS | Geographical Information System |
| IoT | Internet of Things |
| CV | Coefficient of variation |
| $R^2$ | Coefficient of determination |
| BATs | Best available technologies |
| CTD | Conventional tailings disposal |
| TTD | Thickened tailings disposal |
| PTD | Paste tailings disposal |
| FTD | Filtered tailings disposal |
| Cw | Slurry tailings solids content by weight |
| mtpd | Metric tons per day |
| masl | Meters above sea level |
| Ha | Hectare |

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
