# Peer review of "Spatial and Temporal Study of Supernatant Process Water Pond in Tailings Storage Facilities: Use of Remote Sensing Techniques for Preventing Mine Tailings Dam Failures"

_sustainability, doi:10.3390/su15064984_

Round 1

Reviewer 2 Report

1. This paper tried to exploit the spatial and temporal variation of supernatant process water pond in tailings storage facilities by using the techniques of satellite images from remote sensor with GIS, GEE and the application of NDWI, NDVI, mNDWI, EVI. This research design is very useful for monitoring the spatial and temporal variation of large-area water pond or mine tailings beach. 

2. The supplementary analysis about relationship between area and elevation for six locations is suggested because this relationship could affect significantly the variation rate of water pond or mine tailings beach.

3. Both of the outlet velocity of pipeline and elevation difference between discharge point and mine tailings beach decide the length and width of mine tailing beach so that these parameters with operation management need to be added and analyzed.

4. The grain size of mine tailings is the key parameter to beach slope, repose angle, so that it is better to analyze the relationship between grain size and variation rate of mine tailings beach.     

5. The statement of “mNDWI > EVI y EVI <0.1” in Figure 20 could be revised as “mNDWI > EVI & EVI <0.1”.

Reviewer 3 Report

Review Report

Manuscript

Study of the Spatial and Temporal Dynamic Behavior of Super- 2 natant Process Water Pond in Tailings Storage Facilities: Use of 3 Remote Sensing Techniques with Google Earth Engine Tool for 4 Prevent Mine Tailings Dam Failures

Manuscript # sustainability 2242449

Submitted in the journal “Sustainability.

Summary

Mine tailing dam monitoring is becoming an important area for the environmental concerns, personal safety, and economy. Therefore, the prevention of tailings storage facility (TSF) is becoming increasingly important. This study focuses on the spatial temporal behavior of supernatant process water pond and wet tailings in the TSF. Mainly, remote sensing techniques based on multispectral satellite images. These images were analyzed for the deposition, failure criteria, and management of these facilities.

This manuscript needs serious attention before its publication, following are the comments.

Comments

·       First, title need to be revised, as the title is too long, considering rephrasing the title.

·       Novelty has not been explained clearly and in comparison, with the previous studies, there are several studies of similar topic considering the similar approaches for the analysis of imaginary data.

·       Objective to be represent in the form of bullets.

·       Too much random information has been gathered in this manuscript, which can be deleted and thus reducing the overall length of manuscript.

·       Abbreviations/ full form have not been introduced at the first instances in the manuscript, which distract the readers e-g; NDWI (normalized difference water index), mNDVI, NDVI Normalized difference vegetation index, EVI Enhanced vegetation index.

·       Overall length of the manuscript can be shortened, as there are several instance of repetitive text. Table will be helpful in shortening the overall length of the manuscript.

·       The Manuscript is too wordy, it can be shortened with the nice presentation (use of tables, bullets, figures)

·       Repetition of same text repeatedly in section 2.1, all the information can be easily summarized in a table indicating the type, dam name, dam capacity and other relevant information (failure date and damages), Picture can be collage.

·       Line 233 to 240 can be explained in the form of table or bullets.

·       All the affecting parameters can be explained in bullets instead of explaining each point separately and repeating the same text.

·       line 300 to line 307 like line 288 to line 296 (explain the difference or elsewise delete or represent the texted data in form of a table)

·       References is missing in line 442

·       Too long sentences have been found at several sections of manuscript e-g; Line 18 to line 22, line 445 to line 45, Line 748 to line 752, Line 764 to line 770

·       Figure 1, labeling is missing indicating the supernatant process water pond.

·       Supernatant process water pond can be abbreviated.

Suggestions

·       Consider shortening of the long headings.

·       line 96 to line 117 can be deleted or represented using a flow diagram be specific to the topic (readers are not interested in understanding the transportation of slurries)

·       Line 145 to line 166 can be shortened by deleting irrelevant infomation

·       Figure 4 is mis leading as it simultaneously shows the natural terrain and also excavated road level (natural terrain should be dotted)

·       Figure 5, zone to be defined using dotted lines.

·       Figure 9 connection with the present study is not clear

·       Section 2.1.1 to section 2.1.6 can be shortened using a table categorizing the type of tailing dams

·       Reference 61 need attention (should be an English reference)

·       References of Figure 1, 3, 9, 10, 11 are missing.

·       Captioning of figures and table is highly needed to be re-written explaining the proper function and working of figure.

·       Consider revising Figure 2, labelled with incident rays and reflected ray (as this the base of this manuscript)

·       Line 96 to Line 99 can be delete thus references 34, 35 and 14 and 20.

·       Figure 20 does not indicate a complete cycle; the connections are missing (step 1 to step2) similarly for the followed steps.

·        

This study reviews the utilization of robots in the monitoring of infrastructures. It presents the qualitative content analysis of 269 papers on the adoption of robots for inspection. Various forms of robots available and used in construction or infrastructure monitoring have been thoroughly explained concussing previous knowledge on a single point. This manuscript is very well written and explained with strong arguments. However, before this manuscript's publication, the comments need to be incorporated into the final version.

Comments

·       Self-citation has been found in many instances i-e; ref # 20, 124, 126, 127, 180.

·       Thorough check should be carried out for the appropriate cited location of all references, too many references, with repetition, are distracting the interest of the reader. Repetition of the same reference at the end of each sentence has been found in many instances in this manuscript (e-g; lines 346 to 350, line 350 to line 353, line 359 to line 363, line 379 to line 381, Line 650 to 658, line 841 to line 845, line 56 to line 60, line 200 to line 203, line 274 to line 279, at the end of each sentence same reference has been inserted repeatedly without inserting any newly cited reference). One time citation is enough at the end of the complete cited text.

·       Line 148: what does it indicate “in this area”? Are authors pointing towards Robotics and infrastructure monitoring?

·       Line 140 reference is needed "Aria and Cuccullo"

·       Unable to understand Lines: 140 to Line 142, Line 151 to line 152.

·       Line 155 to line 156, average year-on-year growth, seems to be exaggerated a 43 % is too much, Data in Figure 2 indicates very a smaller number of papers in years 1993 to 2012. Please check this thoroughly.

·       line 864 to line 866 reference 251 is wrongly cited it should be 252.

Suggestions 

·       Reference 1 seems to be inappropriate at the cited text location, it can be deleted. Similarly, reference 5 is inappropriate as the said point, the reviewer was unable to find the said point in the cited reference (i-e ref # 5). Therefore, the bibliography may be revised by deleting these references. Also, check for other references as well. 

·       There is no need for citing reference # 3 as the primary source of information is reference # 4 (which is supporting the said argument). Similarly, also check for other instances.

·       Reference # 29 needs to be checked whether it’s the same reference as the authors want to cite.

·       Most of the references cited in the text are not appropriate (reference 90 and 91)

·       Typo mistakes: Line 671, Line 66-67.

Round 2

Reviewer 2 Report

This paper has revised. I have no more comment.

Reviewer 3 Report

Before the publication of this manuscript, some minor modifications can significantly help in improving the quality of this manuscript.

Comments

·       Even though the authors explained novelty, but still too long sentence causes confusion to the readers, so it is suggested to break it into simple lines.

·       It is preferable to reduce the overall length of the manuscript.

·       It seems that authors are not mainly from the Mining Engineering or Mine tailing monitoring field, which is why they think irrelevant information is important for readers. The researchers associated with mine tailing already know the general terminologies.
